# QSVD: Efficient Low-rank Approximation for Unified Query-Key-Value Weight Compression in Low-Precision Vision-Language Models

**Yutong Wang**[1][*]   **Haiyu Wang**[1][*]    **Sai Qian Zhang**[1,2]

[1]Tandon School of Engineering, New York University
[2]Courant Institute of Mathematical Sciences, New York University
{yw6594, hw3689, sai.zhang}@nyu.edu

## Abstract

Vision-Language Models (VLMs) are integral to tasks such as image captioning and visual question answering, but their high computational cost, driven by large memory footprints and processing time, limits their scalability and real-time applicability. In this work, we propose leveraging Singular-Value Decomposition (SVD) over the joint query (Q), key (K), and value (V) weight matrices to reduce KV cache size and computational overhead. We in addition introduce an efficient rank allocation strategy that dynamically adjusts the SVD rank based on its impact on VLM accuracy, achieving a significant reduction in both memory usage and computational cost. Finally, we extend this approach by applying quantization to both VLM weights and activations, resulting in a highly efficient VLM. Our method outperforms previous approaches that rely solely on quantization or SVD by achieving more than $10\%$ accuracy improvement while consuming less hardware cost, making it better for real-time deployment on resource-constrained devices. We open source our code at `https://github.com/SAI-Lab-NYU/QSVD`.

## 1 Introduction

Vision-Language Models (VLMs) are crucial for advancing artificial intelligence by bridging the gap between visual perception and natural language understanding. By enabling machines to interpret and generate both visual and textual information, VLMs open up a wide range of applications, such as image captioning [63, 17, 8, 11], visual question answering [7, 3, 47], and content-based search [22, 43]. These models are vital for tasks where visual context is needed to fully understand textual queries or vice versa, such as healthcare [34, 3, 18], education [61], and interactive entertainment [44, 26].

Despite their strong performance, Vision-Language Models (VLMs) incur substantial computational costs due to the intensive processing required to integrate high-dimensional visual and textual data. Additionally, their autoregressive token generation places significant pressure on memory bandwidth, becoming a major bottleneck for inference speed. To enable practical deployment in latency-sensitive and resource-constrained environments, it is essential to reduce both the computational overhead and the size of the Key-Value (KV) cache, without compromising model accuracy.

To address this issue, particularly the high memory usage introduced by Multi-Head Attention (MHA), several variants have been proposed, such as Grouped-Query Attention [1] and Multi-Query Attention [42, 1], which aim to reduce the number of KV projections while maintaining performance. A recent proposal, Multi-Head Latent Attention (MLA) in the DeepSeek-v3 model [31], offers a novel approach to improving VLM efficiency. It significantly reduces the KV cache size by compressing the KV cache into a latent vector, thereby enhancing inference efficiency.

---

[*]Authors contributed equally; the order of authorship was assigned randomly.

39th Conference on Neural Information Processing Systems (NeurIPS 2025).

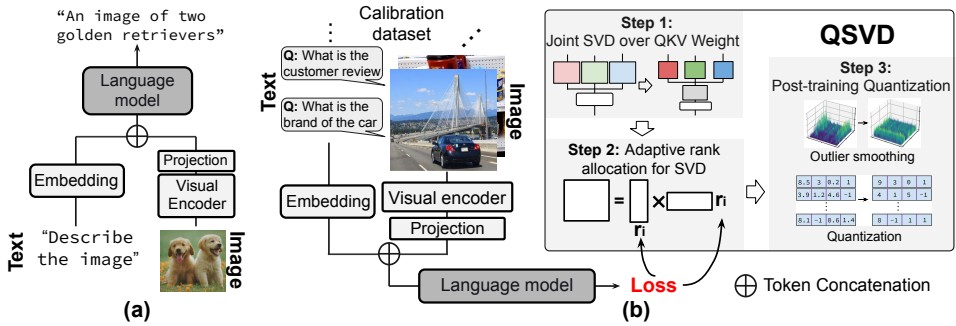

Figure 1: (a) An example on vision-language model. (b) An overview of QSVD.

Building on the insights from MLA, this work proposes the application of Singular-Value Decomposition (SVD), which has proven effective in reducing both KV cache size and computational cost in prior research [51, 60, 52, 28, 25, 5, 49], to the joint weight matrices of the query, key, and value. This approach significantly reduces the KV cache size by storing only the latent vectors instead of separate key and value vectors. Additionally, we introduce a novel rank allocation scheme, which investigates the importance of each singular value in relation to VLM accuracy. This results in a minimal SVD rank with the minimal impact on the model accuracy. Finally, we extend this approach by applying quantization to both VLM weights and activations. The proposed framework, termed *QSVD*, results in an extremely efficient VLM that outperforms previous methods that relied solely on quantization or SVD. Specifically, our contribution can be summarized as follows:

- QSVD proposes applying singular value decomposition to the combined weight matrices of the query, key, and value projections. This technique substantially reduces the size of the KV cache, computational overhead, and weight storage, resulting in significant improvements in hardware efficiency.

- To improve the accuracy of SVD-based compression in VLMs, QSVD proposes a novel importance scoring method that quantifies each singular value's contribution to overall model performance, allowing for rank-based truncation that minimizes accuracy degradation.

- Quantization is applied alongside SVD decomposition to both the weights and activations of the VLM. We propose an efficient method to eliminate outliers under the SVD framework, enabling low-precision operation that reduces the memory footprint of both the KV cache and model weights, while incurring minimal impact on accuracy[2].

## 2 Background and Related Work

### 2.1 Vision Language Model

Vision-Language Models (VLMs) [24, 23, 33, 4, 13, 48] extend the capabilities of Large Language Models (LLMs) by incorporating visual inputs alongside text, enabling tasks such as visual question answering (VQA) and image captioning. Models such as BLIP and InstructBLIP [24, 23] employ data filtering and visual instruction tuning to better align model outputs with human preferences in zero-shot settings. A commonly used architecture, illustrated in Figure 1 (a), processes input images through a visual encoder to produce visual tokens, which are then concatenated with text tokens and passed to a language model for response generation. This concatenation-based design is adopted by widely used models including the LLaVA series [33], SmolVLM [39], PaLI-Gemma [4], and Qwen-VL [48]. Although VLMs demonstrate impressive capabilities, their large size presents challenges in terms of computational efficiency and deployment, particularly on resource-constrained devices. This has led to the development of lightweight alternatives. TinyGPT-V [59] and TinyLLaVA [62] explore efficient designs at smaller scales, while SmolVLM [39] introduces a family of compact models (500M and 2B parameters) that maintain strong performance with much reduced hardware cost.

---

[2]All experiments and data processing were conducted at New York University.

## 2.2 Singular Value Decomposition for Large Models

Singular Value Decomposition (SVD) [19] is a widely used matrix factorization technique that decomposes a matrix $W \in \mathbb{R}^{m \times n}$ into three components: $W = U\Sigma V^T$, where $U$ and $V$ are orthogonal matrices containing the left and right singular vectors of $W$, and $\Sigma$ is a diagonal matrix of non-negative singular values arranged in descending order. By retaining only the top $r$ singular values and corresponding vectors, we obtain a rank-$r$ approximation:

$$W \approx U_r \Sigma_r V_r^T \tag{1}$$

with $U_r \in \mathbb{R}^{m \times r}$, $\Sigma_r \in \mathbb{R}^{r \times r}$, and $V_r \in \mathbb{R}^{n \times r}$. Equivalently, the approximation can be expressed as $W \approx AB$ by defining $A = U_r \Sigma_r^{\frac{1}{2}}$ $B = \Sigma_r^{1/2} V_r^T$ and $B = \Sigma_r^{\frac{1}{2}} V_r^T$. Such low-rank factorizations preserve the most salient structure of $W$ while reducing its dimensionality, enabling matrix compression and accelerating downstream computations.

SVD has been extensively studied as a compression method for LLMs [51, 60, 52, 28, 25, 5, 49]. Early efforts [40] directly applies standard SVD to weight matrices but encountered significant compression errors. To address this, FWSVD [16] prioritizes parameters based on Fisher information [37], while ASVD [60] incorporates activation outliers into the factorization. SVD-LLM [52] further reduces compression loss by explicitly minimizing the contribution of each truncated singular value. Most of these methods focus on compressing model weights. In contrast, Palu [6] and [58] have shifted attention to compressing the KV-Cache, leveraging SVD and low-rank projections to reduce memory footprint. Recent advances include AdaSVD [28], which adaptively compensates for truncation errors and dynamically allocates compression rates according to layer importance, and SVD-LLM V2 [51], which further optimizes singular value truncation via theoretical loss estimation.

Recently, DeepSeek introduces Multi-Head Latent Attention [31], a novel mechanism that integrates low-rank projections directly into the attention computation. Instead of computing attention over the full key and value matrices, this approach projects them into a lower-dimensional latent space using learned projection matrices, effectively reducing the computational and memory costs of multi-head attention without significantly impacting model performance. This latent factorization can be viewed as an implicit low-rank approximation applied dynamically during inference, offering complementary benefits to static weight compression methods such as SVD.

## 2.3 Quantization for Large Models

Post-training quantization (PTQ) has become one of the most used approaches for enabling efficient inference of large models [38, 55, 2, 29, 12, 41, 30, 57, 45, 53, 20, 9]. For example, AffineQuant [38] replaces the traditional scaling factor with a learned affine transformation to better align weight with the quantization grid.

Another line of work focuses on smoothing outliers in activation distributions, which have shown that activations in LLMs contain severe outliers at the per-channel level [2, 9, 29, 35, 30, 55], resulting in substantial quantization errors during activation quantization. To address this issue, SmoothQuant [55] reduces activation outliers by shifting part of the activation outliers into the weights, promoting more balanced quantization. Building upon these ideas, techniques like QuaRot [2], DuQuant [29], and SpinQuant [35] incorporate orthogonal transformations to further enhance quantization performance. These transformations maintain computational invariance by preserving the model's output while effectively suppressing outliers. Specifically, let $W$ and $X$ represent the weight and activation matrices, respectively, where $X$ exhibits channelwise outliers, and let $Y = XW$ denote the output. To eliminate outliers in $X$, an orthogonal matrix $H$ is introduced, satisfying $H^\top H = HH^\top = I$. This transformation yields an equivalent formulation $Y = XW = X'W'$, where $W' = H^\top W$ and $X' = XH$. To minimize runtime overhead, $W' = H^\top W$ can be precomputed offline, and $X' = XH$ can be efficiently integrated into the weight matrices of the previous layer, incurring no additional computational cost. The resulting transformed activation $X'$ exhibits a smoother distribution with significantly fewer outliers, thus lowering the quantization errors. In parallel, methods such as GPTQ [12], OmniQuant [41], and AWQ [30] focus on optimizing scaling factors and channel-wise equalization during the calibration process.

In the realm of VLMs, quantization presents unique challenges due to the integration of visual and textual modalities. QSLAW [56] introduces a quantization-aware scale learning method with a multimodal warmup strategy that progressively incorporates linguistic and multimodal samples

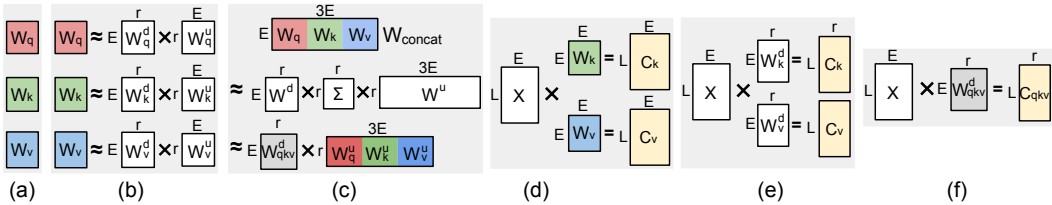

Figure 2: Efficiency analysis of different SVD schemes. **(a)(b)** are original Q/K/V matrix applied SVD. **(c)(d)(e)(f)** are proposed concatenated QKV SVD and their corresponding computing process.

to stabilize training. It also emphasizes group-wise scaling to better handle activation outliers. Q-VLM [46] addresses cross-layer dependency in quantization by leveraging activation entropy as a proxy to guide block partitioning. It formulates a quantization strategy that balances discretization error and search cost, and further optimizes the visual encoder to disentangle cross-layer interactions, enabling more efficient calibration. MBQ [27] proposes a modality-balanced quantization approach that accounts for the distinct gradient distributions of visual and textual tokens during calibration. It applies a modality-aware loss to improve the accuracy of scaling factor estimation. However, to the best of our knowledge, no prior work has combined quantization with SVD for efficient VLM processing in the manner proposed by QSVD.

## 3 Methodology

An overview of QSVD is shown in Figure 1 (b), comprising three key components: joint SVD over the combined QKV weights (Section 3.1), adaptive singular value truncation (Section 3.2), and PTQ over low-rank VLMs (Section 3.3).

### 3.1 Singular-Value Decomposition over Joint QKV Weights

We introduce an efficient SVD-based approach to reduce computation within the multi-head self-attention block, as illustrated in Figure 2, where each subfigure denotes: **(a)** Original QKV matrix in a vision-language model (VLM) without SVD. **(b)** Applying SVD separately to the weight matrices of Q, K, and V, where each of $W_q$, $W_k$, and $W_v$ is factorized into a down- and up-projection pair. **(c)** Our proposed approach: concatenating QKV weights before applying SVD. **(d)** Standard KV computation during prefilling: the input $X$ is multiplied by $W_k$ and $W_v$, and the resulting $C_k$ and $C_v$ are stored in memory. **(e)** Computation with per-matrix SVD: during prefilling, $X$ must be read from memory and multiplied with the down-projection matrices $W_k^d$ and $W_v^d$ to generate low-rank representation of $K$ and $V$. **(f)** Storage and computation in QSVD: since $W_q$, $W_k$, and $W_v$ share a common down-projection matrix $W_{qkv}^d$, $X$ is multiplied by $W_{qkv}^d$ once to produce the intermediate $C_{qkv}$, which is stored and later used to reconstruct the KV vectors. Let $\alpha$, $\eta$, and $\gamma$ denote the weight parameter size, KV cache size, and the computational cost in FLOPs of QKV multiplication, respectively. In the original design, assuming a single-head attention module for simplicity, the combined weight matrices $W_q$, $W_k$, and $W_v$ collectively contain a total of $\alpha_{fp} = 3E^2$ parameters, where $E$ represents the embedding dimension (Figure 2 (a)). The corresponding KV cache requires a memory footprint of $\eta_{fp} = 2LE$, where $L$ is the input sequence length (Figure 2 (d)). The total computational cost in FLOPs for generating the key, query, and value vectors is $\gamma_{fp} = 3LE^2$, where three matrix multiplications are required, each with a size of $(L \times E)$ and $(E \times E)$.

QSVD adopts a more efficient strategy that reduces the number of weight parameters, KV cache size, and overall computational cost. Specifically, the weight matrices $W_Q$, $W_K$, and $W_V$, each of size $E \times E$, are concatenated into a single matrix $W_{\text{concat}} \in \mathbb{R}^{E \times 3E}$. A low-rank SVD is then applied to this combined matrix to achieve compression.

$$[W_q, \ W_k, \ W_v] = W_{concat} \approx W_r^d \times \Sigma_r \times W_r^u \tag{2}$$

$$W_{qkv}^d = W_r^d \Sigma_r^\beta, \ W_{qkv}^u = [W_q^u, \ W_k^u, \ W_v^u] = \Sigma_r^{1-\beta} W_r^u \tag{3}$$

$$[W_q, \ W_k, \ W_v] = W_{qkv}^d \times [W_q^u, \ W_k^u, \ W_v^u] \tag{4}$$

where $W_{qkv}^d \in \mathbb{R}^{E \times r}$, $W_q^u$, $W_k^u$, $W_v^u \in \mathbb{R}^{r \times E}$, and $\beta$ satisfies $0 \leq \beta \leq 1$. After decomposition, the QKV components share a common down-projection matrix $W_{qkv}^d$, while each maintains its

own distinct up-projection matrix. This results in a total weight size of $\alpha_{qsvd} = 4rE$ (Figure 2 (c)). The computational cost for generating the query, key, and value vectors is $\gamma_{qsvd} = 4LrE$, which arises from two steps: first, multiplying the input $X$ with $W_{qkv}^d$ to generate $C_{qkv}$, and then performing a second multiplication with the concatenated matrices $[W_q^u, W_k^u, W_v^u]$. During inference, the intermediate products $C_{qkv}$ between the input and the down-projection matrix $W_{qkv}^d$ are buffered to compute the KV vectors, yielding a total buffer size of $\eta_{qsvd} = rL$ (Figure 2 (f)), and the KV vectors can be easily recomputed as:

$$K = C_{qkv}W_k^u, \; V = C_{qkv}W_v^u \tag{5}$$

In comparison, our method achieves reduced weight size and computational cost when $4rE < 3E^2$ and $4LrE < 3LE^2$, which holds when $r < 0.75E$. This condition is easily satisfied with negligible accuracy loss, as demonstrated by the evaluation results in Section 4. Furthermore, our method consistently reduces the buffered size for the intermediate data, since $rE$ is always smaller than $2E^2$ given that $r < E$. During decoding, the cached intermediate representation $C_{qkv}$ is used to reconstruct the key and value matrices via $W_k^u$ and $W_v^u$, which are then combined with the current query (sequence length $l=1$) to compute the attention outputs.

Previous methods [60, 52] apply SVD individually to the weight matrices, as illustrated in Figure 2(b), resulting in a total of $\alpha_{ind} = 6rE$ parameters. During inference, the intermediate products $C_k$ and $C_v$, which is computed from the input $X$ and the down-projection matrices $W_k^d$ and $W_v^d$, must be buffered, leading to a total buffer size of $\eta_{ind} = 2rL$ (Figure 2(e)). The computational cost for generating the query, key, and value vectors is $\gamma_{ind} = 6LrE$, and the buffer size for $C_k$ and $C_v$ is consistently larger than that required to store the unified $C_{qkv}$ in our method. Finally, our method always achieves a lower weight size, computational cost and intermediate storage.

## 3.2 Cross-layer Rank Allocation for Low-rank SVD

Performing SVD on the joint QKV weights can lead to hardware efficiency gains in both computation and storage, provided that the rank $r$ of $W_{qkv}^d$ is sufficiently reduced without compromising the final accuracy performance. A key challenge, therefore, is determining how to truncate the singular values across all self-attention blocks in the VLM. While prior work has used Fisher information [37] to assess the importance of individual weight matrix or a group of singular values [6, 16], QSVD proposes a more effective method that evaluates the importance of each singular value in a way that minimizes degradation in model accuracy.

Given the SVD of a weight matrix $W = U\Sigma V^T$, it can also be expressed as a summation: $W = \sum_{i=1}^{n} u_i\sigma_i v_i^T$, where $\sigma_i$ is the $i$-th singular value, and $u_i$, $v_i$ are the corresponding left and right singular vectors. Truncating a singular value by setting $\sigma_i = 0$ effectively removes its associated single-rank component from the matrix, resulting in a modified representation $W'_{\sigma_i}$ of $W$, we have:

$$\Delta W_{\sigma_i} = W - W'_{\sigma_i} = u_i\sigma_i v_i^T \tag{6}$$

The truncation of $\sigma_i$ will affect the final output of the VLM and lead to an increase in the training loss $L_t$. The corresponding change in training loss can be estimated through first-order expansion:

$$L_t(W'_{\sigma_i}) = L_t(W - \Delta W_{\sigma_i}) \approx L_t(W) - \sum_{j,k} \Delta W_{\sigma_i}[j,k] \cdot \frac{\partial L_t}{\partial W[j,k]} \tag{7}$$

where $L_t(W'_{\sigma_i})$ denotes the training loss after the weight matrix is modified to $W'_{\sigma_i}$, $W_{\sigma_i}[j,k]$ represents the $(j,k)$-th element of the matrix $W_{\sigma_i}$. Let $G_W$ represent the gradient of the loss with respect to the original weight matrix $W$, the changes on the training loss can be expressed as follows:

$$\Delta L_{\sigma_i} = L_t(W) - L_t(W'_{\sigma_i}) \approx \sum_{j,k} \Delta W_{\sigma_i}[j,k] \cdot G_W[j,k] = \langle \Delta W_{\sigma_i}, G_W \rangle_F \tag{8}$$

where $\langle \cdot, \cdot \rangle_F$ denotes the Frobenius inner product over matrix elements. This formulation enables estimation of each singular value's contribution (e.g., $\sigma_i$) to the change in training loss, providing a principled basis for rank selection by measuring the sensitivity of the loss function to each truncated component across all layers, which can be used to evaluate the importance for each singular value. Specifically, by evaluating $\Delta L_{\sigma_i}$ across multiple calibration samples and computing its squared expectation, we derive the *Importance Score* $\hat{I}_{\sigma_i}$ for each singular value $\sigma_i$, which serves as an

empirical approximation of the diagonal Fisher information:

$$\hat{I}_{\sigma_i} = \mathbb{E}_{x \sim \mathcal{D}}\left[\left(\Delta L_{\sigma_i}^{(n)}\right)^2\right] \approx \frac{1}{N}\sum_{n=1}^{N}\left(\sum_{j,k}\Delta W_{\sigma_i}[j,k]\cdot G_W^{(n)}[j,k]\right)^2 \tag{9}$$

where $\mathcal{D}$ denotes the calibration dataset, $n$ indexes individual samples, and $N$ is the total number of samples in $\mathcal{D}$. However, computing the importance score as defined in Equation 9 poses a significant memory burden, primarily due to the need to construct and store $\Delta W_{\sigma_i}[j,k]$ for all singular values. Since each $\Delta W_{\sigma_i}$ is a full $E \times E$ matrix and there are $E$ such singular values from the joint SVD, the total memory cost scales as $\mathcal{O}(E^3)$ per layer, making naive computation impractical for large models. To address this, the importance score $\hat{I}_{\sigma_i}$ can alternatively be computed as follows:

$$\hat{I}_{\sigma_i} = \frac{1}{N}\sum_{n=1}^{N}\sigma_i^2\left[U^T G_W^{(n)}V\right]_{(i,i)}^2 \tag{10}$$

where $U$ and $V$ are the left and right singular vectors from the SVD, $\sigma_i$ is the $i$-th singular value. The notation $(i,i)$ refers to the $i$-th diagonal element of the transformed gradient matrix $U^T G_W^{(n)}V$. The proof is given in Appendix A.1. This form eliminates the need to compute and store $\Delta W_{\sigma_i}$ for each singular value, requiring only $\mathcal{O}(E^2)$ memory instead of $\mathcal{O}(E^3)$ per layer.

The overall SVD procedure is as follows. Starting with the original model, we first concatenate the QKV weight matrices and apply joint SVD, following the method outlined in Section 3.1. We adopt the activation-aware SVD technique from ASVD [60] to extract the singular values across all layers. For each singular value, we then compute its corresponding importance score based on the calibration dataset, as defined in Equation 10. After computing the importance scores, we perform cross-layer ranking by globally sorting all singular values based on their scores. We retain only the top $k$ singular values with the highest importance scores, where $k$ termed *rank budget*. All remaining singular values are truncated. This global ranking strategy ensures that the most critical components are preserved regardless of the layer they originate from, allowing for an optimal allocation of rank capacity throughout the VLM. For QSVD, we apply the rank selection mechanism to the self-attention layers throughout the entire VLM.

### 3.3 Post-Training Quantization Scheme for Low-Rank VLMs

Building on the efficient low-rank SVD approach described in Section 3.1 and Section 3.2, this section presents an efficient quantization scheme applied to the resulting low-rank VLMs for further hardware efficiency enhancement. To analyze the outlier distribution in the VLM, we profile the input activation distribution of LLaVA-v1.5 13B [33]. Specifically, we examine the input activations $X$ across the self-attention modules and feed-forward modules within the language model of the

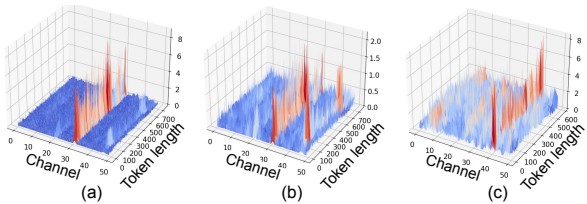

Figure 3: Input activation distribution within VLM. Only partial channel are shown.

VLM, as illustrated in Figure 3 (a) and (b), respectively. Our analysis reveals prominent channelwise outliers in $X$ across all three components, which poses the great challenge when quantizing these VLM activations for low-precision operations.

To address this issue, we adopt the rotational method introduced in [2, 54] and outlined in Section 2.3 to smooth channelwise outliers. However, since the self-attention architecture of the VLM has been modified by the application of SVD, we develop an efficient quantization approach that accounts for this change. Let $X$ denote the input to the weight matrices, and the output be expressed as $Y = XW_{qkv}^d W_{qkv}^u = C_{qkv}W_{\text{qkv}}^u$, where $W_{qkv}^d = W_r^d\Sigma_r^\beta$ and $W_{qkv}^u = \Sigma_r^{1-\beta}W_r^u$, as notated in Equation 3. The distribution of $C_{qkv}$, which is buffered for KV vector recomputation, is shown in Figure 3 (c). We observe that $C_{qkv}$ exhibits channelwise outliers, rendering it unsuitable for low-precision quantization. To eliminate the channelwise outliers in both $X$ and the compressed representation $C_{qkv}$, we introduce orthogonal matrices $H_1$ and $H_2$. The self-attention computation together with its quantized counterpart are then reformulated as follows:

$$Y = (XH_1^\top)(H_1 W_{qkv}^d H_2^\top)(H_2 W_{qkv}^u) \qquad Y' = Q(C_{qkv})Q(H_2 W_{qkv}^u) \tag{11}$$

Table 1: Accuracy evaluation of different methods. For ASVD and SVDLLM, their $R_1, R_2$ are shared. Detailed results can be found in Appendix A.2.

| | Method | ScienceQA-IMG ↑ | | | | | | | | VizWiz ↑ | | | |
|---|---|---|---|---|---|---|---|---|---|---|---|---|---|
| | | Acc. | Hw cost | Acc. | Hw cost | Acc. | Hw cost | Acc. | Hw cost | Acc. | Hw cost | Acc. | Hw cost |
| SmolVLM 2B | ASVD | 53.84% | $R_1:100\%$ | 7.88% | $R_1:90.0\%$ | 0.69% | $R_1:80.0\%$ | 0.10% | $R_1:70.0\%$ | 6.68% | $R_1:100\%$ | 0.00% | $R_1:80.0\%$ |
| | SVDLLM | 65.89% | $R_2:50.0\%$ | 34.61% | $R_2:42.5\%$ | 9.07% | $R_2:35.0\%$ | 3.02% | $R_2:27.5\%$ | 14.86% | $R_2:50.0\%$ | 0.13% | $R_2:35.0\%$ |
| | **QSVD-noQ** | **83.78%** | $R_1:100\%$ $R_2:37.5\%$ | **81.70%** | $R_1:90.0\%$ $R_2:33.75\%$ | **79.57%** | $R_1:80.0\%$ $R_2:30.0\%$ | **77.64%** | $R_1:70.0\%$ $R_2:26.25\%$ | **40.67%** | $R_1:100\%$ $R_2:37.5\%$ | **40.67%** | $R_1:80.0\%$ $R_2:30.0\%$ |
| | FP16 | Accuracy: 84.53% | | | | | | | | Accuracy: 37.07% | | | |
| LLaVA-Next 7B | ASVD | 50.72% | $R_1:63.3\%$ | 47.15% | $R_1:60.0\%$ | 40.26% | $R_1:56.7\%$ | 25.73% | $R_1:53.3\%$ | 47.78% | $R_1:63.3\%$ | 39.41% | $R_1:56.7\%$ |
| | SVDLLM | 65.94% | $R_2:22.5\%$ | 66.14% | $R_2:20.0\%$ | 64.90% | $R_2:17.5\%$ | 62.87% | $R_2:15.0\%$ | 48.01% | $R_2:22.5\%$ | 47.74% | $R_2:17.5\%$ |
| | **QSVD-noQ** | **69.91%** | $R_1:60.0\%$ $R_2:22.5\%$ | **68.22%** | $R_1:53.3\%$ $R_2:20.0\%$ | **67.03%** | $R_1:46.7\%$ $R_2:17.5\%$ | **65.15%** | $R_1:40.0\%$ $R_2:15.0\%$ | **54.38%** | $R_1:60.0\%$ $R_2:22.5\%$ | **51.42%** | $R_1:46.7\%$ $R_2:17.5\%$ |
| | FP16 | Accuracy: 69.51% | | | | | | | | Accuracy: 54.46% | | | |
| LLaVA-v1.5 13B | ASVD | 64.70% | $R_1:63.3\%$ | 56.92% | $R_1:60.0\%$ | 46.50% | $R_1:56.7\%$ | 42.79% | $R_1:53.3\%$ | 44.48% | $R_1:63.3\%$ | 40.01% | $R_1:56.7\%$ |
| | SVDLLM | 71.44% | $R_2:22.5\%$ | 71.44% | $R_2:20.0\%$ | 71.29% | $R_2:17.5\%$ | 70.50% | $R_2:15.0\%$ | 51.03% | $R_2:22.5\%$ | 49.37% | $R_2:17.5\%$ |
| | **QSVD-noQ** | **71.79%** | $R_1:60.0\%$ $R_2:22.5\%$ | **71.74%** | $R_1:53.3\%$ $R_2:20.0\%$ | **71.74%** | $R_1:46.7\%$ $R_2:17.5\%$ | **70.80%** | $R_1:40.0\%$ $R_2:15.0\%$ | **56.15%** | $R_1:60.0\%$ $R_2:22.5\%$ | **55.79%** | $R_1:46.7\%$ $R_2:17.5\%$ |
| | FP16 | Accuracy: 71.78% | | | | | | | | Accuracy: 53.63% | | | |

where $C_{qkv}$ can be approximately computed using the quantized input and weight as:

$$C_{qkv} \approx Q(XH_1^\top)Q(H_1W_{qkv}^dH_2^\top) \tag{12}$$

The computations presented in Equation 11 and Equation 12 support low-precision execution while reducing the size of both the weight parameters and the intermediate results $C_{qkv}$, thereby lowering the overall memory footprint and reduce the processing latency.

Although the introduction of $H_1$ and $H_2$ helps mitigate outliers in $X$ and $C_{qkv}$, respectively, we observe that these transformations do not fully eliminate the severe outliers, particularly those present in $C_{qkv}$. To analyze this issue, we examine the distribution of $W_{qkv}^d$, which directly influences the distribution of $C_{qkv}$. We observe that $W_{qkv}^d = W_r^d\Sigma_r^\beta$ is strongly affected by the parameter $\beta$. Since $\Sigma_r$ is a diagonal matrix whose entries are singular values that can vary significantly in magnitude, raising them to the power $\beta$ can amplify the disparity. This, in turn, exacerbates the presence of outliers in $C_{qkv}$, as shown by the following derivation:

$$C_{qkv} = XW_{qkv}^d = XW_r^d\Sigma_r^\beta = XW_r^d diag(\sigma_1^\beta, \sigma_2^\beta, ..., \sigma_r^\beta) = [\sigma_1^\beta(XW_r^d)_1, ..., \sigma_r^\beta(XW_r^d)_r] \tag{13}$$

where $(XW_r^d)_i$ denotes the $i$-th column of $XW_r^d$, which can significantly influence the channelwise outlier distribution in $C_{qkv}$. To address this, we propose learning an optimal value for $\beta$ by optimizing it over the calibration dataset $\mathcal{D}$, namely:

$$\min_\beta \sum_{d \in D} ||Y_d - Y_d'||^2 \tag{14}$$

where $Y_d$ and $Y_d'$ denote the self-attention block outputs with and without quantization for the $d$-th sample in the calibration dataset $\mathcal{D}$, respectively. The parameter $\beta$ is optimized individually for each layer within the VLM. Finally, we apply the quantization operations to both the visual encoder and all layers of the language model, resulting in an end-to-end efficient VLM computation.

## 4 Evaluation Results

We evaluate QSVD on five VLMs: LLaVA-v1.5 7B [33], LLaVA-v1.5 13B, LLaVA-Next 7B, LLaVA-Next 13B, and SmolVLM-Instruct [39]. To determine the optimal rank allocation and $\beta$ parameters, we use 256 samples from the ScienceQA training dataset [36], following the procedures outlined in Section 3.2 and Section 3.3. For evaluation, we adopt three widely used benchmark datasets, ScienceQA [36], VizWiz [15], and SEED-Bench-IMG [21], in line with prior work such as LLaVA. We compare QSVD against baseline methods by implementing them on the aforementioned VLM models, the baselines include the SVD approaches (ASVD [60], SVD-LLM [52]) and quantization approach (QuaRot [2], DuQuant [29], QVLM [46]). Specifically, for ASVD and SVD-LLM, we follow their official implementations by applying SVD separately to the Key and Value matrices, while avoiding decomposition of the Query matrices to prevent performance degradation. Additionally, SVD is not applied to other linear layers within the VLM. All methods are evaluated using the same calibration samples and random seeds to ensure fairness, and we report their best performance. For QuaRot and DuQuant, we apply them to the various VLMs by strictly following the detailed procedures provided in their respective code repositories.

Table 2: Quantization evaluation across different models and datasets. $R_1$ is omitted since they are similar for different methods. Detailed results can be found in Appendix A.2.

| Model | Bit | Duquant [29] | | | | QVLM [46] | | | | QASVD | | | | Ours | | | |
|---|---|---|---|---|---|---|---|---|---|---|---|---|---|---|---|---|---|
| | | $R_2$ | SciQA↑ | VizWiz↑ | SEED↑ | $R_2$ | SciQA↑ | VizWiz↑ | SEED↑ | $R_2$ | SciQA↑ | VizWiz↑ | SEED↑ | $R_2$ | SciQA↑ | VizWiz↑ | SEED↑ |
| LLaVA-v1.5 7B | FP16 | 100% | 68.01% | 50.03% | 60.18% | 100% | 68.01% | 50.03% | 60.18% | 100% | 68.01% | 50.03% | 60.18% | 100% | 68.01% | 50.03% | 60.18% |
| | W8A8 | 50% | 66.53% | 49.86% | 58.62% | 50% | 64.65% | 50.64% | 51.82% | 50% | 52.95% | 48.31% | 53.92% | 18.75% | 67.57% | 54.06% | 60.20% |
| | W8A4 | 25% | 57.36% | 50.07% | 54.11% | 25% | 55.24% | 48.33% | 50.13% | 25% | 41.92% | 47.85% | 41.26% | 9.38% | 65.61% | 52.18% | 58.49% |
| | W4A4 | 25% | 52.56% | 48.77% | 49.50% | 25% | 51.12% | 47.38% | 34.00% | 25% | 12.61% | 1.23% | 10.48% | 9.38% | 55.16% | 52.05% | 52.69% |
| LLaVA-v1.5 13B | FP16 | 100% | 71.78% | 53.63% | 62.53% | 100% | 71.78% | 53.63% | 62.53% | 100% | 71.78% | 53.63% | 62.53% | 100% | 71.78% | 53.63% | 62.53% |
| | W8A8 | 50% | 69.66% | 50.73% | 62.70% | 50% | 70.65% | 50.32% | 62.36% | 50% | 70.25% | 54.93% | 61.84% | 18.75% | 72.12% | 55.42% | 62.91% |
| | W8A4 | 25% | 67.22% | 53.07% | 61.43% | 25% | 66.46% | 49.03% | 59.22% | 25% | 65.34% | 52.61% | 59.30% | 9.38% | 70.12% | 53.20% | 62.95% |
| | W4A4 | 25% | 65.80% | 49.37% | 59.28% | 25% | 64.86% | 48.57% | 41.07% | 25% | 20.35% | 37.5% | 20.96% | 9.38% | 65.82% | 56.82% | 61.79% |
| LLaVA-Next 7B | FP16 | 100% | 69.60% | 54.46% | 69.02% | 100% | 69.60% | 54.46% | 69.02% | 100% | 69.60% | 54.46% | 69.02% | 100% | 69.60% | 54.46% | 69.02% |
| | W8A8 | 50% | 66.34% | 52.05% | 67.91% | 50% | 64.70% | 47.55% | 66.82% | 50% | 64.94% | 47.3% | 66.87% | 18.75% | 69.09% | 53.42% | 68.92% |
| | W8A4 | 25% | **66.34%** | 50.26% | 63.64% | 25% | 60.60% | 48.55% | 50.38% | 25% | 43.37% | 48.65% | 49.63% | 9.38% | 66.10% | 53.72% | 65.63% |
| | W4A4 | 25% | 58.37% | 54.26% | **62.95%** | 25% | 55.30% | 48.58% | 45.24% | 25% | 19.17% | 3.30% | 13.68% | 9.38% | 59.67% | 52.00% | 62.08% |
| LLaVA-Next 13B | FP16 | 100% | 73.23% | 57.72% | 71.30% | 100% | 73.23% | 57.72% | 71.30% | 100% | 73.23% | 57.72% | 71.30% | 100% | 73.23% | 57.72% | 71.30% |
| | W8A8 | 50% | 61.13% | 54.38% | 70.07% | 50% | 69.86% | 49.89% | 69.28% | 50% | 71.52% | 55.13% | 67.87% | 18.75% | 72.38% | 58.33% | 71.23% |
| | W8A4 | 25% | 70.20% | 52.43% | 66.15% | 25% | 65.28% | 48.98% | 65.39% | 25% | 64.85% | 53.13% | 66.54% | 9.38% | 70.43% | 58.52% | 69.21% |
| | W4A4 | 25% | 58.16% | 53.26% | 63.15% | 25% | 57.33% | 52.23% | 60.55% | 25% | 12.85% | 4.44% | 14.64% | 9.38% | 63.61% | 54.27% | 65.34% |
| Smol VLM 2B | FP16 | 100% | 84.68% | 37.07% | 68.18% | 100% | 84.68% | 37.07% | 68.18% | 100% | 84.68% | 37.07% | 68.18% | 100% | 84.68% | 37.07% | 68.18% |
| | W8A8 | 50% | 57.80% | 35.52% | 62.65% | 50% | 55.30% | 33.14% | 55.73% | 50% | 11.94% | 0.00% | 9.13% | 18.75% | 76.20% | 37.82% | 66.49% |
| | W8A4 | 25% | 55.92% | 34.09% | 45.82% | 25% | 53.52% | 30.43% | 42.24% | 25% | 3.92% | 0.00% | 1.23% | 9.38% | 62.35% | 36.91% | 55.35% |

Given that QSVD performs joint SVD with quantization, we first evaluate its SVD-only performance in by comparing it with ASVD and SVD-LLM under equivalent hardware costs, including intermediate data storage for KV recomputation, weight size, and VLM computational cost in FLOPs. Specifically, we express it in terms of the ratio between ours and the FP16 without compression. We denote the SVD-only approach depicted in Section 4.1 as **QSVD-noQ**. We then apply the quantization techniques introduced in Section 3.3 to the SVD-compressed VLM and compare the results with advanced quantization methods such as DuQuant [29] and QVLM [46], as well as quantized version of ASVD (QASVD). For QASVD, we apply QuaRot [2] to the SVD-truncated VLMs obtained from ASVD. The corresponding results are presented in Section 4.2.

We evaluate all performance results on NVIDIA RTX A6000 GPUs using `VLMEvalKit` [10], and we report results under three weight-activation quantization configurations: W8A8 (8-bit weights and 8-bit activations), W8A4, and W4A4. For activation quantization, we apply per-token symmetric quantization. For weight quantization, we use round-to-nearest (RTN) with per-channel symmetric quantization and a learnable clipping ratio, determined via linear search to minimize squared error, following [2]. We present ablation studies in Section 4.3 and evaluate the latency improvements of QSVD on GPU in Section 4.4. Additional results are included in Appendix A.2.

## 4.1 Accuracy Evaluation on QSVD-noQ

We begin by evaluating the QSVD-noQ performance in FP16 under four different rank budgets $k$. To ensure a fair comparison, we adjust the rank configurations of all methods such that our approach consistently maintains the lowest hardware cost in terms of intermediate data storage ($\eta$), weight size ($\alpha$), and computational overhead ($\gamma$), as mentioned in Section 3.1. Importantly, as noted in Section 3.1, the relative ratios among the weight sizes $\alpha_{fp}$, $\alpha_{qsvd}$, and $\alpha_{ind}$ are identical to the ratios among the computational costs $\gamma_{fp}$, $\gamma_{qsvd}$, and $\gamma_{ind}$. This equivalence allows us to report a single normalized metric to represent both weight parameter reduction and computational efficiency. Therefore we have $R_1$ and $R_2$, defined as:

$$R_1 = \frac{\alpha_i}{\alpha_{fp}} = \frac{\gamma_i}{\gamma_{fp}} \qquad R_2 = \frac{\eta_i}{\eta_{fp}} \tag{15}$$

where $i$ can either be "qsvd" or "ind". The evaluation results are presented in Table 1. Our method outperforms ASVD and SVD-LLM in accuracy while incurring minimal or comparable hardware cost. On LLaVA-v1.5 13B, QSVD-noQ results in less than a $1\%$ drop in ScienceQA-IMG accuracy compared to the original FP16 baseline, and notably even surpasses the FP16 performance on VizWiz. For instance, with $R_1 = 46.7\%$ and $R_2 = 17.5\%$, QSVD-noQ achieves an accuracy of $55.79\%$, exceeding the FP16 counterpart by more than $2\%$. This may be due to the low-rank approximation effectively mitigating hallucinations [32] in the VLM; however, further investigation is needed to confirm this hypothesis. Moreover, our approach consistently achieves higher accuracy than ASVD and SVD-LLM under reduced parameter and cache ratios ($R_1$ and $R_2$), with the performance gap

widening as these ratios decrease. For instance, in the SmolVLM setting, our method attains over 70% accuracy on ScienceQA-IMG, while both ASVD and SVD-LLM fail to operate effectively.

## 4.2 Accuracy Evaluation of QSVD

The low-rank SVD components are subsequently quantized using the techniques described in Section 3.3. We compare QSVD with DuQuant [29] and QVLM [46], as well as QASVD, which integrate QuaRot's quantization approach with the low-rank SVD outputs from ASVD, respectively. Quantization is applied consistently across the entire VLM, including both feed-forward and self-attention layers in the language model and visual encoder. Evaluations are conducted on three benchmark datasets: ScienceQA, VizWiz, and SEEDBench. All methods maintain a similar $R_1$ of approximately 50%, while $R_2$, which has a greater impact on inference latency and cache size, varies across approaches. Notably, QSVD consistently achieves a lower $R_2$ compared to all other baselines.

As shown in Table 2, under the W8A8 setting, QSVD consistently outperforms other baselines in most scenarios. On large-scale models like LLaVA-1.5 13B, it reaches accuracy comparable to the FP16 baseline while reducing QKV weights and compute by 50%, and cutting intermediate data size to just 18.75%. Under the more aggressive W8A4 setting, QSVD surpasses all baselines and approaches FP16-level performance using as little as 9.38% of the original KV cache. Finally, Table 2 shows the quantization results under the W4A4 setting. Under this configuration, QASVD fail to operate properly (yielding zero accuracy). Despite the challenging conditions, QSVD consistently delivers the highest performance among all models while maintaining the lowest hardware cost in terms of $R_1$ and $R_2$.

## 4.3 Ablation Study

**Effectiveness of Cross-layer Rank Allocation Scheme**
To evaluate the effectiveness of our cross-layer rank allocation strategy (Section 3.2), we compare it with two baseline methods. The first, referred to as the Uniform-rank (UR) scheme, applies SVD to the joint QKV weights using the same rank across all VLM blocks. The second, denoted as the Fisher Information-Based (FIB) scheme, also applies SVD to the joint QKV weights but distributes ranks across layers based on Fisher information. This approach has been adopted in prior work for SVD-based compression in LLMs [5]. All methods operate under the

Table 3: Accuracy performance under varying rank allocation strategies.

| | Method | ScienceQA-IMG ↑ | | | |
|---|---|---|---|---|---|
| | FP16 | 71.78% | | | |
| LLaVA-v1.5-13B | $R_1$ | 60.0% | 53.3% | 46.7% | 40.0% |
| | $R_2$ | 22.5% | 20.0% | 17.5% | 15.0% |
| | UR | **71.84%** | 70.40% | 70.40% | 67.72% |
| | FIB | 70.60% | 70.60% | 70.15% | 69.96% |
| | **QSVD-noQ** | 71.79% | **71.74%** | **71.74%** | **70.80%** |

same hardware budget, defined by $R_1$ and $R_2$. As shown in Table 3, under aggressive compression, QSVD-noQ consistently outperforms both baselines and maintains accuracy close to the FP16 model.

**Impact of Learning $\beta$**  As described in Equation 14, we train $\beta$ to suppress outliers in the intermediate result $C_{qkv}$. Table 4 presents the impact of the learnable $\beta$ on VLM accuracy under W4A4 setting over LLaVA 7Bs on Science QA. We compare QSVD with baseline methods using a fixed $\beta$ across the entire VLM. QSVD consistently achieves the highest accuracy, outperforming all fixed-$\beta$ baselines, highlighting the importance of learning $\beta$ for effective low-bit quantization.

Table 4: Impact of $\beta$ learning.

| Model | 0.0 | **QSVD** | 0.4 | 0.8 |
|---|---|---|---|---|
| v1.5-7b | 54.53% | **55.16%** | 54.83% | 6.09% |
| Next-7b | 58.80% | **59.67%** | 56.56% | 15.12% |

**Long Sequence Scenarios**  To further evaluate the adaptability of our QSVD method under long sequence conditions, we conduct experiments on the HRBench-4K dataset [50], which consists of 4K-resolution images. We follow the same evaluation setup as mentioned above and use `VLMEvalKit` [10] to report the "Average All" accuracy metric. Both LLaVA-Next 13B and LLaVA-v1.5 13B are evalu-

Table 5: Evaluation results on HRBench-4K.

| | Method | HRBench-4K ↑ | | | | | |
|---|---|---|---|---|---|---|---|
| | | Acc. | Hw cost | Acc. | Hw cost | Acc. | Hw cost |
| LLaVA-Next 13B | ASVD | 44.38% | $R_1$ :63.3% $R_2$ :22.5% | 44.12% | $R_1$ :60.0% $R_2$ :20.0% | 43.12% | $R_1$ :56.7% $R_2$ :17.5% |
| | **QSVD-noQ** | **44.88%** | $R_1$ :**60.0%** $R_2$ :**22.5%** | **44.12%** | $R_1$ :**53.3%** $R_2$ :**20.0%** | **43.88%** | $R_1$ :**46.7%** $R_2$ :**17.5%** |
| | FP16 | Accuracy: 45.63% | | | | | |
| LLaVA-v1.5 13B | ASVD | 39.12% | $R_1$ :63.3% $R_2$ :22.5% | 38.62% | $R_1$ :60.0% $R_2$ :20.0% | 36.62% | $R_1$ :56.7% $R_2$ :17.5% |
| | **QSVD-noQ** | **39.88%** | $R_1$ :**60.0%** $R_2$ :**22.5%** | **38.75%** | $R_1$ :**53.3%** $R_2$ :**20.0%** | **39.00%** | $R_1$ :**46.7%** $R_2$ :**17.5%** |
| | FP16 | Accuracy: 39.12% | | | | | |

Table 6: Accuracy evaluation results (↑) on HallusionBench under different compressed parameter size ratios ($R_1$). FP16 indicates uncompressed original models.

| $R_1$ | LLaVA-v1.5 13B | | | | LLaVA-Next 13B | | | |
|---|---|---|---|---|---|---|---|---|
| | aAcc | fAcc | qAcc | Overall | aAcc | fAcc | qAcc | Overall |
| 90% | 49.63% | 21.10% | 17.58% | 29.44% | 57.73% | 26.01% | 26.59% | 36.78% |
| 80% | 48.90% | 20.52% | 16.92% | 28.78% | 58.25% | 26.01% | 26.81% | 37.03% |
| 70% | **50.26%** | **22.83%** | **17.80%** | **30.30%** | **58.46%** | **26.88%** | **27.25%** | **37.53%** |
| FP16 | 44.69% | 19.36% | 16.04% | 26.70% | 56.78% | 26.01% | 25.27% | 36.02% |

ated under our QSVD-noQ configuration and compared against ASVD and FP16 baselines. The results are summarized in Table 5.

As shown in Table 5, QSVD-noQ consistently outperforms ASVD in all evaluation settings. Moreover, the relative performance trends on HRBench-4K closely mirror those observed on ScienceQA-IMG and VizWiz, indicating that our rank allocation strategy generalizes effectively to long sequence scenarios resulting from high-resolution visual inputs.

**Impact of QSVD on Hallucination** We further evaluate the impact of QSVD on VLM hallucination using HallusionBench [14], following the same evaluation setup as mentioned above. Metrics include aAcc, fAcc, and qAcc from HallusionBench and their overall average score. As shown in Table 6, both LLaVA-v1.5 13B and LLaVA-Next 13B exhibit noticeable improvements in groundedness metrics after QSVD-noQ. For LLaVA-Next 13B, the overall score increases from 36.02 to a peak of 37.53 at $R_1 = 70\%$. Similarly, LLaVA-v1.5 13B improves from 26.70 to 30.30 at $R_1 = 70\%$, marking a clear reduction in hallucination. These findings confirm that QSVD-noQ not only reduces model and cache size but also acts as an effective regularizer against hallucinations. This effect explains why, on certain benchmark datasets such as VizWiz, the QSVD-compressed models occasionally outperform their original FP16 counterparts in terms of end-task accuracy.

## 4.4 Latency Evaluation on VLM

QSVD leverages both SVD and quantization to jointly compress model weights and KV cache, making it well-suited for deployment on memory-constrained hardware. We evaluate inference latency of the layer-wise LLaVA-v1.5 7B on an NVIDIA RTX 4070 GPU with 12GB memory. The batch size is set to 1 and the token length to 4K. As shown in Figure 4 , under FP16 precision, due to limited GPU memory, both the FP16 baseline and QSVD-noQ require partial offloading to CPU memory. However, QSVD-noQ with $40\%$ and $30\%$ (denoted as noQ-40% and noQ-30%) rank retention benefits from reduced data movement enabled by effective SVD compression, achieving up to a $2.1\times$ speedup over the baseline. Furthermore, QSVD with W8A8 quantization, under $70\%$ and $50\%$ rank retention, completely avoids offloading and achieves a significant speedup of up to $13.1\times$.

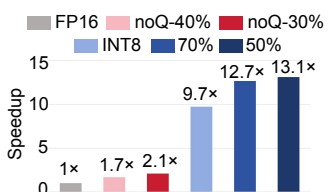

Figure 4: Normalized speedup of QSVD-noQ and QSVD W8A8 on low-end GPU.

## 5 Conclusion and Limitation

In this work, we proposed QSVD, a unified framework that applies joint singular value decomposition and quantization to compress VLMs efficiently. By decomposing the combined QKV weight matrices and introducing an adaptive cross-layer rank allocation strategy, QSVD significantly reduces computational cost, KV cache size, and model storage with minimal impact on accuracy. Although quantization is applied to all layers of the VLM, compression is mainly focused on the QKV weights in self-attention layers. Future work will explore joint optimization across all model blocks. Additionally, improving VLM efficiency may also make powerful models more accessible, which raises concerns about potential misuse in areas such as surveillance, misinformation, and privacy violations. Further investigation is needed to address these risks.

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

# A Technical Appendices and Supplementary Material

**Outline of Appendices**

> **A.1:** Detailed derivation for Importance Score in Sec. 3.2.
>
> **A.2:** Additional experimental results.
>
> **A.3:** Case study for QSVD

## A.1 Importance Score derivation

**Proof.** Recall that the singular value decomposition (SVD) of $W$ is given by:

$$W = U\Sigma V^T = \sum_{i=1}^{r} \sigma_i u_i v_i^T$$

where $u_i$ and $v_i$ are the $i$-th left and right singular vectors, and $\sigma_i$ is the $i$-th singular value. When truncating $\sigma_i$, the change in $W$ is:

$$\Delta W_{\sigma_i} = \sigma_i u_i v_i^T$$

Then, the inner product between $\Delta W_i$ and the gradient $G_W^{(n)}$ is:

$$\sum_{j,k} \Delta W_{\sigma_i}(j,k) \cdot G_W^{(n)}(j,k) = \langle \Delta W_{\sigma_i}, G_W^{(n)} \rangle_F$$

where $\langle \cdot, \ \cdot \rangle_F$ denotes the Frobenius inner product.

Substitute $\Delta W_{\sigma_i} = \sigma_i u_i v_i^T$:

$$\langle \Delta W_{\sigma_i}, G_W^{(n)} \rangle_F = \langle \sigma_i u_i v_i^T, \ G_W^{(n)} \rangle_F = \sigma_i \langle u_i v_i^T, \ G_W^{(n)} \rangle_F$$

Using the property of the Frobenius inner product:

$$\langle A, \ B \rangle_F = \text{tr}(A^T B)$$

we have:

$$\sigma_i \langle u_i v_i^T, \ G_W^{(n)} \rangle_F = \sigma_i \, \text{tr}((u_i v_i^T)^T G_W^{(n)}) = \sigma_i \, \text{tr}(v_i u_i^T G_W^{(n)})$$

By cyclic property of trace:

$$\sigma_i \, \text{tr}(v_i u_i^T G_W^{(n)}) = \sigma_i \, \text{tr}(u_i^T G_W^{(n)} v_i)$$

Since $u_i^T G_W^{(n)} v_i$ is a scalar:

$$\sigma_i \, \text{tr}(u_i^T G_W^{(n)} v_i) = \sigma_i (u_i^T G_W^{(n)} v_i)$$

Note that:

$$U^T G_W^{(n)} V \in \mathbb{R}^{r \times r}$$

and the $(i, i)$-th diagonal entry is:

$$[U^T G_W^{(n)} V]_{(i,i)} = u_i^T G_W^{(n)} v_i$$

Therefore:

$$\sum_{j,k} \Delta W_{\sigma_i}(j,k) \cdot G_W^{(n)}(j,k) = \sigma_i [U^T G_W^{(n)} V]_{(i,i)}$$

$$\sum_{j,k} \Delta W_{\sigma_i}(j,k) \cdot G_W(j,k) = \langle \Delta W_{\sigma_i}, G_W \rangle_F = \sigma_i [U^T G_W V]_{(i,i)}$$

Finally, the importance score $\hat{I}_{\sigma_i}$ can be computed as follows:

$$\hat{I}_{\sigma_i} = \frac{1}{N} \sum_{n=1}^{N} \sigma_i^2 \left[ U^T G_W^{(n)} V \right]_{(i,i)}^2$$

If a whitening transformation such as ASVD or SVD-LLM is applied prior to SVD, that is, $U\Sigma V^T = \text{SVD}(WS)$ where $S$ denotes the whitening matrix, the corresponding importance score can be reformulated as:

$$\hat{I}_{\sigma_i} = \frac{1}{N} \sum_{n=1}^{N} \sigma_i^2 \left[ U^T G_W^{(n)} S^{-T} V \right]_{(i,i)}^2$$

where the term $S^{-T}$ converts the gradient from the original weight space to the whitened space in which the SVD is performed.

## A.2    Full table for experiments

Here we include more detailed experiments table for QSVD method. We evaluate all performance reuslts on NVIDIA RTX A6000 GPUs, we report results under QSVD-noQ and under three weight-activation quantization configurations: W8A8 (8-bit weights and 8-bit activations), W8A4, and W4A4. For activation quantization, we apply per-token symmetric quantization. For weight quantization, we use round-to-nearest (RTN) with per-channel symmetric quantization and a learnable clipping ratio, determined via linear search to minimize squared error, following [2]. For QSVD baseline, we add QuaRot without SVD as an baseline, also for ours method, we use activation clip ratio of 0.85 for vit model and 0.9 for language model, under this setting, we have updated some QSVD accuracy results higher than main paper report.

**QSVD-noQ results.** Table 7 presents detailed results of QSVD-noQ on SmolVLM [39], LLaVA-v1.5 [33] series, and LLaVA-Next series, along with the corresponding preserved ratios. QSVD-noQ consistently outperforms ASVD and SVD-LLM in accuracy under reduced parameter and cache ratios ($R_1$ and $R_2$), with the performance gap widening as the compression becomes more aggressive. For instance, in the SmolVLM setting, our method maintains over $70\%$ accuracy on ScienceQA-IMG [36], while both ASVD [60] and SVD-LLM [51] fail to function effectively.

**QSVD Results.** We evaluate our proposed QSVD quantization strategy on LLaVA v1.5 and Next series: 7B, 13B, across three benchmarks: ScienceQA, VizWiz, and SEEDBench. Table 8 summarizes performance under two low-bit settings, W8A8 and W8A4.

Under the W8A8 setting, QSVD matches or exceeds prior methods such as Duquant [29], QVLM [46], and QASVD, while reducing the KV cache and intermediate data sizes by up to 50%. Compared to QSVDLLM, our approach avoids the need for manually re-optimizing decomposed matrices while still achieving superior performance.

In the more challenging W8A4 configuration, QSVD continues to deliver robust outputs, reaching levels comparable to the FP16 baseline using just $9.38\%$ of the original KV cache. This demonstrates the scalability of our quantization design under aggressive memory constraints.

| | Method | ScienceQA-IMG ↑ | | | | VizWiz ↑ | | | |
|---|---|---|---|---|---|---|---|---|---|
| **SmolVLM-2B** | FP16 | 84.53% | | | | 37.07% | | | |
| | $R_1$ | 100% | 90.0% | 80.0% | 70.0% | 100% | 90.0% | 80.0% | 70.0% |
| | $R_2$ | 50.0% | 42.5% | 35.0% | 27.5% | 50.0% | 42.5% | 35.0% | 27.5% |
| | ASVD | 53.84% | 7.88% | 0.69% | 0.10% | 6.68% | 0.00% | 0.00% | 0.00% |
| | SVDLLM | 65.89% | 34.61% | 9.07% | 3.02% | 14.86% | 1.62% | 0.13% | 0.00% |
| | $R_1$ | **100%** | **90.0%** | **80.0%** | **70.0%** | **100%** | **90.0%** | **80.0%** | **70.0%** |
| | $R_2$ | **37.5%** | **33.75%** | **30.0%** | **26.25%** | **37.5%** | **33.75%** | **30.0%** | **26.25%** |
| | **QSVD-noQ** | **83.78%** | **81.70%** | **79.57%** | **77.64%** | **40.67%** | **39.88%** | **40.67%** | **43.84%** |
| **LLaVA-v1.5 7B** | FP16 | 68.01% | | | | 50.03% | | | |
| | $R_1$ | 63.3% | 60.0% | 56.7% | 53.3% | 63.3% | 60.0% | 56.7% | 53.3% |
| | $R_2$ | 22.5% | 20.0% | 17.5% | 15.0% | 22.5% | 20.0% | 17.5% | 15.0% |
| | ASVD | 22.36% | 19.09% | 15.22% | 10.81% | 47.70% | 39.02% | 10.10% | 8.87% |
| | SVDLLM | 55.23% | 55.03% | 49.23% | 51.17% | 50.10% | 50.71% | 50.47% | 49.99% |
| | $R_1$ | **60.0%** | **53.3%** | **46.7%** | **40.0%** | **60.0%** | **53.3%** | **46.7%** | **40.0%** |
| | $R_2$ | **22.5%** | **20.0%** | **17.5%** | **15.0%** | **22.5%** | **20.0%** | **17.5%** | **15.0%** |
| | **QSVD-noQ** | **66.12%** | **65.64%** | **64.06%** | **61.68%** | **53.84%** | **54.19%** | **53.88%** | **52.40%** |
| **LLaVA-v1.5-13B** | FP16 | 71.78% | | | | 53.63% | | | |
| | $R_1$ | 63.3% | 60.0% | 56.7% | 53.3% | 63.3% | 60.0% | 56.7% | 53.3% |
| | $R_2$ | 22.5% | 20.0% | 17.5% | 15.0% | 22.5% | 20.0% | 17.5% | 15.0% |
| | ASVD | 64.70% | 56.92% | 46.50% | 42.79% | 44.48% | 40.63% | 40.01% | 37.87% |
| | SVDLLM | 71.44% | 71.44% | 71.29% | 70.50% | 51.03% | 51.15% | 49.37% | 46.49% |
| | $R_1$ | **60.0%** | **53.3%** | **46.7%** | **40.0%** | **60.0%** | **53.3%** | **46.7%** | **40.0%** |
| | $R_2$ | **22.5%** | **20.0%** | **17.5%** | **15.0%** | **22.5%** | **20.0%** | **17.5%** | **15.0%** |
| | **QSVD-noQ** | **71.79%** | **71.74%** | **71.74%** | **70.80%** | **56.15%** | **56.05%** | **55.79%** | **54.04%** |
| **LLaVA-Next 7B** | FP16 | 69.51% | | | | 54.46% | | | |
| | $R_1$ | 63.3% | 60.0% | 56.7% | 53.3% | 63.3% | 60.0% | 56.7% | 53.3% |
| | $R_2$ | 22.5% | 20.0% | 17.5% | 15.0% | 22.5% | 20.0% | 17.5% | 15.0% |
| | ASVD | 50.72% | 47.15% | 40.26% | 25.73% | 47.78% | 47.3% | 39.41% | 6.69% |
| | SVDLLM | 65.94% | 66.14% | 64.90% | 62.87% | 48.01% | 48.41% | 47.74% | 47.73% |
| | $R_1$ | **60.0%** | **53.3%** | **46.7%** | **40.0%** | **60.0%** | **53.3%** | **46.7%** | **40.0%** |
| | $R_2$ | **22.5%** | **20.0%** | **17.5%** | **15.0%** | **22.5%** | **20.0%** | **17.5%** | **15.0%** |
| | **QSVD-noQ** | **69.91%** | **68.22%** | **67.03%** | **65.15%** | **54.38%** | **52.31%** | **51.42%** | **49.86%** |
| **LLaVA-Next-13B** | FP16 | 73.23% | | | | 57.72% | | | |
| | $R_1$ | 63.3% | 60.0% | 56.7% | 53.3% | 63.3% | 60.0% | 56.7% | 53.3% |
| | $R_2$ | 22.5% | 20.0% | 17.5% | 15.0% | 22.5% | 20.0% | 17.5% | 15.0% |
| | ASVD | 69.71% | 68.86% | 67.43% | 64.01% | 55.42% | 54.97% | 54.50% | 52.95% |
| | SVDLLM | 70.30% | 69.71% | 69.56% | 68.52% | 53.08% | 52.54% | 52.52% | 51.77% |
| | $R_1$ | **60.0%** | **53.3%** | **46.7%** | **40.0%** | **60.0%** | **53.3%** | **46.7%** | **40.0%** |
| | $R_2$ | **22.5%** | **20.0%** | **17.5%** | **15.0%** | **22.5%** | **20.0%** | **17.5%** | **15.0%** |
| | **QSVD-noQ** | **72.63%** | **72.29%** | **72.34%** | **71.64%** | **55.48%** | **55.14%** | **54.99%** | **55.77%** |

Table 7: Accuracy on ScienceQA-IMG and VizWiz datasets. The $R_1, R_2$ denotes the proportion of preserved QKV parameters and the corresponding cache ratio.

For completeness, Appendix Tables 8 and 9 report full results across all model variants and bitwidth configurations. Notably, our method consistently ranks highest or near-highest across settings, while maintaining favorable compression ratios of $50\%/18.75\%$ in W8A8 and $50\%/9.38\%$ in W8A4. These results highlight the ability of QSVD to balance compression and output quality across a diverse range of architectures and tasks.

| | Method | W8A8 | | | $R_1/R_2$ | W8A4 | | | $R_1/R_2$ |
|---|---|---|---|---|---|---|---|---|---|
| | | SciQA↑ | VizWiz↑ | SEED↑ | | SciQA↑ | VizWiz↑ | SEED↑ | |
| **LLaVA-1.5-7B** | FP16 | 68.01% | 50.03% | 60.18% | 100%/100% | 68.01% | 50.03% | 60.18% | 100%/100% |
| | QuaRot | **67.90%** | 49.95% | 60.11% | 50%/50% | 63.19% | 49.82% | 58.18% | 50%/25% |
| | Duquant | 66.53% | 49.86% | 58.62% | 50.52%/50% | 57.36% | 50.07% | 54.11% | 50.52%/25% |
| | QVLM | 64.65% | 50.64% | 51.82% | 50%/50% | 55.24% | 48.33% | 50.13% | 50%/25% |
| | QASVD | 52.95% | 48.31% | 53.92% | 50%/50% | 41.92% | 47.85% | 41.26% | 50%/25% |
| | QSVDLLM | 66.14% | 51.93% | 56.47% | 50%/50% | 30.38% | 45.00% | 37.00% | 50%/25% |
| | **QSVD** | 67.57% | **54.06%** | **60.20%** | 50%/18.75% | 65.61% | 52.18% | 58.49% | 50%/9.38% |
| **LLaVA-1.5-13B** | FP16 | 71.80% | 53.63% | 62.54% | 100%/100% | 71.80% | 53.63% | 62.54% | 100%/100% |
| | QuaRot | 71.64% | 53.64% | 62.57% | 50%/50% | 68.02% | **54.57%** | 58.53% | 50%/25% |
| | Duquant | 69.66% | 50.73% | 62.70% | 51.67%/50% | 67.22% | 53.07% | 61.43% | 51.67%/25% |
| | QVLM | 70.65% | 50.32% | 62.36% | 50%/50% | 66.46% | 49.03% | 59.22% | 50%/25% |
| | QASVD | 70.25% | 54.93% | 61.84% | 50%/50% | 65.34% | 52.61% | 59.30% | 50%/25% |
| | QSVDLLM | 70.65% | **56.32%** | 62.35% | 50%/50% | 60.20% | 50.52% | 55.03% | 50%/25% |
| | **QSVD** | **72.12%** | 55.42% | **62.91%** | 50%/18.75% | 70.12% | 53.20% | 62.95% | 50%/9.38% |
| **LLaVA-Next-7B** | FP16 | 69.60% | 54.46% | 69.02% | 100%/100% | 69.60% | 54.46% | 69.02% | 100%/100% |
| | QuaRot | **69.19%** | 52.86% | 65.60% | 50%/50% | 64.53% | 51.27% | 65.08% | 50%/25% |
| | Duquant | 66.34% | 52.05% | 67.91% | 50.52%/50% | **66.34%** | 50.26% | 63.64% | 50.52%/25% |
| | QVLM | 64.70% | 47.55% | 66.82% | 50%/50% | 60.60% | 48.55% | 50.38% | 50%/25% |
| | QASVD | 64.94% | 47.30% | 66.87% | 50%/50% | 43.37% | 48.65% | 49.63% | 50%/25% |
| | QSVDLLM | 64.70% | 47.55% | 66.83% | 50%/50% | 33.83% | 46.05% | 39.08% | 50%/25% |
| | **QSVD** | 69.09% | **53.42%** | **68.92%** | 50%/18.75% | 66.10% | **53.72%** | 65.63% | 50%/9.38% |
| **LLaVA-Next-13B** | FP16 | 73.23% | 57.72% | 71.30% | 100%/100% | 73.23% | 57.72% | 71.30% | 100%/100% |
| | QuaRot | 72.04% | 58.03% | 67.29% | 50%/50% | 66.98% | 55.56% | **70.15%** | 50%/25% |
| | Duquant | 61.13% | 54.38% | 70.07% | 51.67%/50% | 70.20% | 52.43% | 66.15% | 51.67%/25% |
| | QVLM | 69.86% | 49.89% | 69.28% | 50%/50% | 65.28% | 48.98% | 65.39% | 50%/25% |
| | QASVD | 71.52% | 55.13% | 67.87% | 50%/50% | 64.85% | 53.13% | 66.54% | 50%/25% |
| | QSVDLLM | 69.85% | 49.89% | 69.27% | 50%/50% | 61.25% | 45.05% | 65.03% | 50%/25% |
| | **QSVD** | **72.38%** | **58.33%** | **71.23%** | 50%/18.75% | 70.43% | 58.52% | 69.21% | 50%/9.38% |

Table 8: Quantization results on W8A8 and W8A4.

| | Bit | Method | LLaVA-V1.5 Series | | | LLaVA-Next Series | | | $R_1/R_2$ |
|---|---|---|---|---|---|---|---|---|---|
| | | | ScienceQA ↑ | SEED ↑ | VizWiz ↑ | ScienceQA ↑ | SEED ↑ | VizWiz ↑ | |
| **7B** | – | FP16 | 68.01% | 60.18% | 50.03% | 69.60% | 69.02% | 54.46% | 100% / 100% |
| | W4A4 | QuaRot | 49.08% | 50.54% | 49.96% | 55.57% | 59.81% | **55.25%** | 25% / 25% |
| | W4A4 | Duquant | 52.56% | 49.51% | 48.77% | 58.36% | 62.95% | 52.00% | 27.08% / 25% |
| | W4A4 | QVLM | 51.12% | 34.00% | 47.38% | 55.30% | 45.24% | 48.58% | 25% / 25% |
| | W4A4 | QASVD | 12.61% | 10.48% | 1.23% | 19.17% | 13.68% | 3.30% | 25% / 25% |
| | W4A4 | QSVDLLM | 6.18% | 5.53% | 0.00% | 10.13% | 8.64% | 2.55% | 25% / 25% |
| | W4A4 | **QSVD** | **55.16%** | **52.70%** | **52.05%** | **59.67%** | **62.97%** | 52.00% | 25% / 9.38% |
| **13B** | – | FP16 | 71.80% | 62.54% | 53.63% | 73.23% | 71.30% | 57.72% | 100% / 100% |
| | W4A4 | QuaRot | 62.74% | 60.14% | 55.62% | 57.47% | 62.95% | 50.13% | 25% / 25% |
| | W4A4 | Duquant | 65.80% | 59.28% | 49.37% | 58.16% | 63.15% | 53.26% | 26.67% / 25% |
| | W4A4 | QVLM | 64.86% | 41.07% | 48.57% | 57.33% | 60.55% | 52.23% | 25% / 25% |
| | W4A4 | QASVD | 20.35% | 37.5% | 20.96% | 12.85% | 14.64% | 4.44% | 25% / 25% |
| | W4A4 | QSVDLLM | 10.53% | 7.65% | 1.01% | 15.57% | 10.89% | 2.22% | 25% / 25% |
| | W4A4 | **QSVD** | **65.82%** | **61.79%** | **56.82%** | **63.61%** | **65.34%** | **54.27%** | 25% / 9.38% |

Table 9: Quantization results on W4A4.

### A.3 Case study for QSVD

For case study, we randomly selected examples from the ScienceQA [36] test set, which demonstrate our method's superior performance over QVLM and QASVD baselines.

- As shown in Case 2 and 6, where the FP16 model fails to produce the correct answers, QSVD exhibits more consistent responses between W8A8 and W4A4 settings, and does not contradict the FP16 outputs.

- QSVD also demonstrates surprisingly better performance under low-bit settings in Case 1 and 7, where the FP16 model answers incorrectly but QSVD at W4A4 produces the correct response. This may be attributed to the quantization process reducing model noise, as discussed in Dobi-SVD [49], and may partially explain why QSVD sometimes surpasses FP16 accuracy, as observed in Appendix A.2.

- In Case 3, 4, 5, and 8, QSVD successfully answers questions that FP16 answers correctly but where other baselines fail under W4A4, highlighting its robustness under extreme quantization.

- Furthermore, in Case 4, 5, and 7, other baselines output random tokens or repeat content under 4-bit settings, while QSVD maintains coherent and relevant responses.

**Case 1: `LLaVA-v1.5-7b`**

**Question:** The diagrams below show two pure samples of gas in identical closed, rigid containers. Each colored ball represents one gas particle. Both samples have the same number of particles. Compare the average kinetic energies of the particles in each sample. Which sample has the higher temperature?

A. sample B
B. neither; the samples have the same temperature
C. sample A

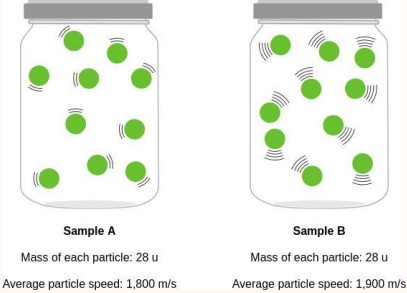

Sample A
Mass of each particle: 28 u
Average particle speed: 1,800 m/s

Sample B
Mass of each particle: 28 u
Average particle speed: 1,900 m/s

**FP16 Response:** 1. Compare the average kinetic energies of the particles in each sample. Sample A: The average kinetic energy of the particles in Sample A is 18000000000000000000000000000 [truncated]

- - - - - - - - - - - - - - - - - - - - - - - - - - - - - - - - - - - - - - - - - - - - - - - - -

**QVLM W8A8 Response:** The image shows two jars filled with green balls, which represent the particles in the gas. The jars are filled with the same number of particles, and the balls are of the same color. The image also provides information about the average kinetic energy of the particles in each jar.

To determine which jar has the higher temperature, we need to compare the average kinetic energy of the particles in each jar. If the particles in jar A have a higher average kinetic energy, then jar A has a higher temperature. If the particles in jar B have a higher average kinetic energy, then jar B has a higher temperature.

However, the image does not provide enough information to determine the average kinetic energy of the particles in each jar. Therefore, we cannot confidently determine which jar has the higher temperature.

**QASVD W8A8 Response:** 1. The diagrams below show two pure samples of gas in identical closed, rigid containers. Each colored ball represents one gas particle. Both samples have the same number of particles. 2. Compare the average kinetic energies of the particles in each sample. Which sample has the higher temperature? A. sample B B. neither; the samples have the same temperature C. sample A

- - - - - - - - - - - - - - - - - - - - - - - - - - - - - - - - - - - - - - - - - - - - - - - - -

**QSVD W8A8 Response:** 1. The diagrams below show two pure samples of gas in identical closed, rigid containers. Each colored ball represents one gas particle. Both samples have the same number of particles. 2. Compare the average kinetic energies of the particles in each sample. Which sample has the higher temperature?
A. sample B B. neither; the samples have the same temperature C. sample A

**QSVD W4A4 Response:** A. sample B

- - - - - - - - - - - - - - - - - - - - - - - - - - - - - - - - - - - - - - - - - - - - - - - - -

**Note that QASVD and QVLM W4A4 output nothing or random number.**

**Case 2:** `LLaVA-v1.5-13b`

**Question:** The diagrams below show two pure samples of gas in identical closed, rigid containers. Each colored ball represents one gas particle. Both samples have the same number of particles. Compare the average kinetic energies of the particles in each sample. Which sample has the higher temperature?

A. sample B
B. neither; the samples have the same temperature
C. sample A

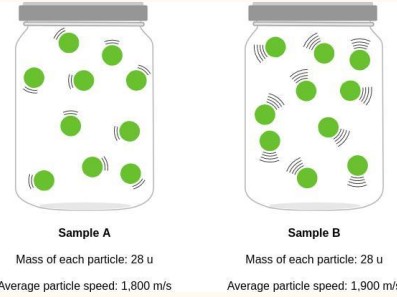

**Sample A**
Mass of each particle: 28 u
Average particle speed: 1,800 m/s

**Sample B**
Mass of each particle: 28 u
Average particle speed: 1,900 m/s

**FP16 Response:** The average kinetic energy of the particles in sample A is higher than that of sample B. This is because the particles in sample A are moving faster than those in sample B. The higher average kinetic energy in sample A indicates that the temperature of the gas in sample A is higher than that of the gas in sample B.

- - - - - - - - - - - - - - - - - - - - - - - - - - - - - - - - - - - - - - - - - - - - - - -

**QVLM W8A8 Response:** The average kinetic energy of the particles in each sample can be determined by calculating the average kinetic energy of each colored ball. The higher the average kinetic energy, the higher the temperature of the sample.
In the image, there are 28 colored balls in each sample. The average kinetic energy of the particles in sample A is calculated as follows:
(1/28) (1/28) (1/28) (1/28) (1/28) (1/28) (1/28) (1/28) (1/28) (1/28) (1/28) (1/28) (1/28)
(1/28) (1/28) (1/28) (1/28) (1/28) (1/28) (1/28) (1/28) (1/28) (1/28) (1/28) (1/28) (1/28)
(1/28) (1/28) (1/28) (1/28) (1/28) (1/28) (1/28) (1/28) (1/28) (1/28) (1/28) (1/28) (1/28)
(1/28) (1/28) (1/28) (1/28) (1/28) (1/28) (1/28) (1/28) (1/28) (1/28) (1/28) (1/28) (1/28)
(1/28) (1/28) (1/

**QASVD W8A8 Response:** The average kinetic energy of the particles in sample A is higher than that of sample B. This means that sample A has a higher temperature than sample B.

- - - - - - - - - - - - - - - - - - - - - - - - - - - - - - - - - - - - - - - - - - - - - - -

**QSVD W8A8 Response:** The diagrams show two samples of gas, each in a closed, rigid container. The containers have the same number of colored balls, which represent the gas particles. The samples have different average kinetic energies, which can be determined by comparing the average kinetic energy of the particles in each sample.
Sample A has a higher average kinetic energy than Sample B. This is because the balls in Sample A are moving faster than the balls in Sample B. The average kinetic energy of the particles in Sample A is higher than that of Sample B, which means that Sample A has a higher temperature.

**QSVD W4A4 Response:** The average kinetic energy of the particles in sample A is higher than that of sample B.

- - - - - - - - - - - - - - - - - - - - - - - - - - - - - - - - - - - - - - - - - - - - - - -

**Note that QASVD and QVLM W4A4 output nothing or random number.**

**Case 3:** `LLaVA-v1.5-13b`

**Question:** What is the capital of Louisiana?
A. Atlanta
B. Frankfort
C. New Orleans
D. Baton Rouge

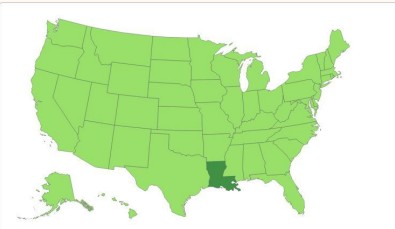

**FP16 Response:** D. Baton Rouge

------------------------------------------------------------

**QVLM W8A8 Response:** C. New Orleans
**QVLM W4A4 Response:** C. New Orleans
**QASVD W8A8 Response:** C. New Orleans

------------------------------------------------------------

**QSVD W8A8 Response:** D. Baton Rouge
**QSVD W4A4 Response:** The capital of Louisiana is D. Baton Rouge.

------------------------------------------------------------

**Note that QASVD W4A4 output a random number.**

---

**Case 4:** `LLaVA-v1.5-13b`

**Question:** Figure: Kermadec Arc. The Kermadec Arc is a deep sea ecosystem in the southern Pacific Ocean. Which better describes the Kermadec Arc ecosystem?
A. It has bright sunlight. It also has organisms that crawl or stick to the ground.
B. It has water at the bottom of the ocean. It also has no sunlight.

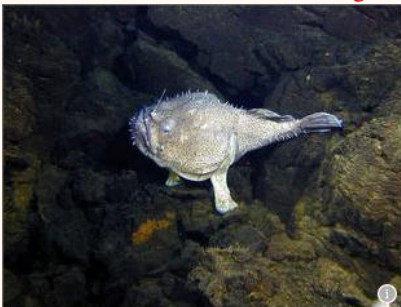

**FP16 Response:** B. It has water at the bottom of the ocean. It also has no sunlight.

------------------------------------------------------------

**QVLM W8A8 Response:** B. It has water at the bottom of the ocean. It also has no sunlight..
**QVLM W4A4 Response:** B. It has water at the bottom of the ocean. It also has no sunlight..
**QASVD W8A8 Response:** B. It has water at the bottom of the ocean. It also has no sunlight..
**QASVD W4A4 Response:** The Kermadec Arc is a deep sea ecosystem in the southern Pacific Ocean. Which better describes the Kermadec Arc ecosystem? B. It has water at the bottom of the ocean. It also has no sunlight.

------------------------------------------------------------

**QSVD W8A8 Response:** B. It has water at the bottom of the ocean. It also has no sunlight.
**QSVD W4A4 Response:** B. It has water at the bottom of the ocean. It also has no sunlight..

------------------------------------------------------------

**Case 5:** `LLaVA-v1.5-13b`

**Question:** The model below represents silver. Silver is a precious metal that is used to make many items, such as jewelry and coins. Complete the statement. Silver is ().
A. an elementary substance
B. a compound

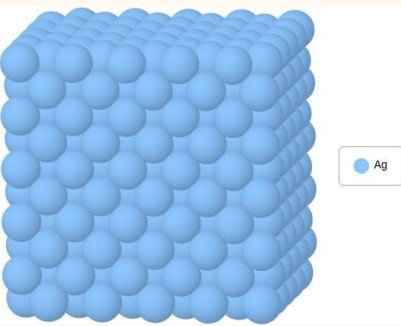

**FP16 Response:** A. an elementary substance

- - - - - - - - - - - - - - - - - - - - - - - - - - - - - - - - - - - - - - - - - - - - - - - - - - - - - -

**QVLM W8A8 Response:** A. an elementary substance
**QVLM W4A4 Response:** Silver is a precious metal that is used to make many items, such as jewelry and coins.
**QASVD W8A8 Response:** A. an elementary substance

- - - - - - - - - - - - - - - - - - - - - - - - - - - - - - - - - - - - - - - - - - - - - - - - - - - - - -

**QSVD W8A8 Response:** A. an elementary substance
**QSVD W4A4 Response:** A. an elementary substance

- - - - - - - - - - - - - - - - - - - - - - - - - - - - - - - - - - - - - - - - - - - - - - - - - - - - - -

**Note that QASVD W4A4 output a random number.**

**Case 6:** `LLaVA-v1.5-13b`

**Question:** The images below show two pairs of magnets. The magnets in different pairs do not affect each other. All the magnets shown are made of the same material. Think about the magnetic force between the magnets in each pair. Which of the following statements is true?
A. The magnetic force is stronger in Pair 1.
B. The strength of the magnetic force is the same in both pairs.
C. The magnetic force is stronger in Pair 2.

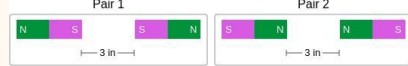

**FP16 Response:** 1. The magnetic force is stronger in Pair 1.
In the image, there are two pairs of magnets, each consisting of two magnets. The magnets in different pairs do not affect each other, and all the magnets are made of the same material. The statement "The magnetic force is stronger in Pair 1" suggests that the magnetic force between the magnets in Pair 1 is stronger than the magnetic force between the magnets in Pair 2. This could be due to various factors such as the orientation, size, or shape of the magnets in each pair. However, without more information or a clear visual comparison, it is not possible to definitively determine which pair has a stronger magnetic force.

- - - - - - - - - - - - - - - - - - - - - - - - - - - - - - - - - - - - - - - - -

**QVLM W8A8 Response:** The magnetic force is stronger in Pair 2.
**QVLM W4A4 Response:** 1. The magnetic force is stronger in Pair 1.
**QASVD W8A8 Response:** The magnetic force is stronger in Pair 2.

- - - - - - - - - - - - - - - - - - - - - - - - - - - - - - - - - - - - - - - - -

**QSVD W8A8 Response:** 1. The magnetic force is stronger in Pair 1.
In the image, there are two pairs of magnets, each consisting of three magnets. The magnets in each pair are arranged in a straight line, and they are made of the same material. The magnetic force between the magnets in each pair is determined by their strength, size, and arrangement. Since the magnets in Pair 1 are larger and more closely spaced, the magnetic force between them is likely to be stronger than the force between the magnets in Pair 2.
**QSVD W4A4 Response:** The magnetic force is stronger in Pair 1.

- - - - - - - - - - - - - - - - - - - - - - - - - - - - - - - - - - - - - - - - -

**Note that QASVD W4A4 output a random number.**

---

**Case 7:** `LLaVA-v1.5-13b`

**Question:** Which ocean is highlighted?
A. the Indian Ocean
B. the Atlantic Ocean
C. the Pacific Ocean
D. the Southern Ocean

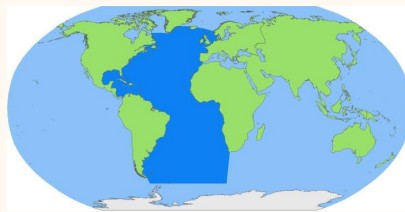

**FP16 Response:** D. the Southern Ocean

- - - - - - - - - - - - - - - - - - - - - - - - - - - - - - - - - - - - - - - - -

**QVLM W8A8 Response:** D. the Southern Ocean
**QVLM W4A4 Response:** the Southern Ocean
**QASVD W8A8 Response:** C. the Pacific Ocean
**QASVD W4A4 Response:** 2. C. the Pacific Ocean

- - - - - - - - - - - - - - - - - - - - - - - - - - - - - - - - - - - - - - - - -

**QSVD W8A8 Response:** D. the Southern Ocean
**QSVD W4A4 Response:** B. the Atlantic Ocean

- - - - - - - - - - - - - - - - - - - - - - - - - - - - - - - - - - - - - - - - -

**Case 8:** `LLaVA-v1.5-13b`

**Question:** This organism is Fromia monilis. It is a member of the animal kingdom. Fromia monilis is commonly called a tile sea star. Did you notice that this tile sea star is missing one of its arms? If a tile sea star loses an arm, it can regrow the arm over time. This ability is called regeneration. Can Fromia monilis cells make their own food?
A. yes
B. no

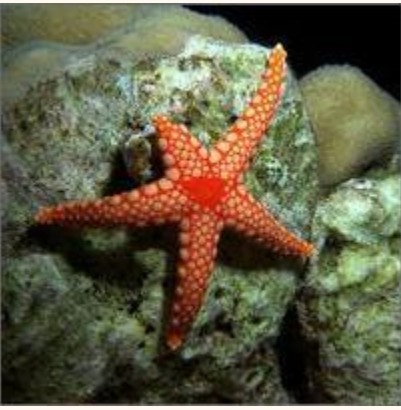

**FP16 Response:** 1. Fromia monilis is a type of sea star, also known as a tile sea star. 2. The sea star is missing one of its arms, which can regrow over time. 3. Fromia monilis cells cannot make their own food , as they are not capable of photosynthesis.

- - - - - - - - - - - - - - - - - - - - - - - - - - - - - - - - - - - - - - - - - - - - - - - -

**QVLM W8A8 Response:** 1. Fromia monilis is a type of sea star, also known as a tile sea star. 2. The sea star is missing one of its arms, which can regrow over time. 3. Fromia monilis cells cannot make their own food, as they are not capable of photosynthesis.

**QVLM W4A4 Response:** 1. Fromia monilis is a type of sea star that belongs to the animal kingdom. 2. Fromia monilis is commonly known as a tile sea star. 3. The tile sea star is missing one of its arms. 4. Fromia monilis can regrow its arms over time, a process called regeneration. 5. Fromia monilis cells can make their own food.

**QASVD W8A8 Response:** Fromia monilis is a type of sea star, which is an invertebrate. Invertebrates are animals that do not have a backbone. They are typically found in aquatic environments, such as oceans and seas.
Sea stars, like Fromia monilis, are known for their ability to regrow lost body parts, such as arms. This process is called regeneration.
Sea stars are not capable of making their own food, as they are primarily herbivores. They feed on various types of algae and other plant materials found in their environment.

**QASVD W4A4 Response:** 1. A. Yes

- - - - - - - - - - - - - - - - - - - - - - - - - - - - - - - - - - - - - - - - - - - - - - - -

**QSVD W8A8 Response:** 1. Fromia monilis is a type of sea star, also known as a tile sea star. 2. The sea star is missing one of its arms, which can regrow over time. 3. Fromia monilis cells cannot make their own food, as they are not capable of photosynthesis.

**QSVD W4A4 Response:** Fromia monilis is a type of sea star that is commonly found in the ocean. It is a member of the animal kingdom, and it is commonly known as a tile sea star. Fromia monilis has the ability to regenerate lost body parts, such as an arm. This ability is called regeneration.In terms of whether Fromia monilis cells can make their own food, the answer is no. Fromia monilis is a carnivorous organism, which means it relies on other organisms for food. It feeds on small marine animals such as crabs, clams, and other invertebrates.

