# OpenReview forum: "QSVD: Efficient Low-rank Approximation for Unified Query-Key-Value Weight Compression in Low-Precision Vision-Language Models"
_NeurIPS.cc/2025/Conference — NeurIPS 2025 spotlight_

### Official Review · Reviewer_HD9e · 2025-07-01

**Clarity:** 3
**Significance:** 2
**Originality:** 2
**Rating:** 4
**Confidence:** 3

**Summary:**

This paper proposes an efficient method for compressing Vision-Language Models (VLMs) by applying Singular Value Decomposition (SVD) to the QKV weight matrices, reducing KV cache size and computation. A dynamic rank allocation strategy is introduced to balance compression and accuracy. The approach is further enhanced with quantization of both weights and activations. Experiments show that the method achieves better accuracy–efficiency trade-offs than prior work, making it suitable for real-time, resource-constrained settings.

**Questions:**

See Weakness

**Ethical Concerns:**

["NO or VERY MINOR ethics concerns only"]

**Final Justification:**

Thank you for the authors’ response. I feel that most of my concerns have been addressed. However, the explanation of how the proposed method specifically tackles the unique challenges of VLMs remains somewhat unclear. Therefore, I have changed my rating from borderline reject to borderline accept.

**Limitations:**

Yes

**Quality:**

2

**Strengths And Weaknesses:**

Strengths:

1.	The paper is well-structured and easy to follow.
2.	The experiments and ablations are thorough and clearly demonstrate the effectiveness of the proposed method.

Weaknesses:

1.	My main concern is that the method appears to lack VLM-specific design—it primarily targets the QKV weights, which is a general approach applicable to LLMs as well. This raises the question: why is the method not directly evaluated on standard LLM benchmarks and compared against existing LLM compression techniques, if it's not tailored to vision-language settings?
2.	The paper lacks comparisons with state-of-the-art post-training quantization methods in terms of inference time, memory usage, and accuracy.
3.	The figures need to be more refined. For example, Figure 2 alone is hard to interpret—the connections and meanings of the different subplots are unclear.

---

> ### Author Rebuttal · Authors · 2025-07-31
>
> Thank you for your insightful comments. Below, we summarize the key concerns and questions raised in your review and provide detailed responses to each point.
>
> **Q1: QSVD primarily targets the QKV weights, which is a general approach applicable to LLMs as well. This raises the question: why is the method not directly evaluated on standard LLM benchmarks and compared against existing LLM compression techniques, if it's not tailored to vision-language settings?**
>
> **Ans:** Efficiency enhancement techniques specifically targeting VLMs have not been studied as extensively as those for language models. Given the growing computational demands of VLM deployment, we believe QSVD is both timely and essential. Our joint QKV compression approach directly addresses the unique efficiency challenges of multimodal inference, setting it apart from conventional unimodal compression methods used in language models.
>
> In addition, whether the proposed techniques can be effectively applied to LLMs requires further investigation. This is because the quantization scheme in QSVD is specifically tailored to the input activation distributions observed in VLMs. As shown in our analysis (Section 3.3, Figure 3), VLM activations display distinct channel-wise outlier patterns. The quantization methods we introduce are designed based on this distribution. Consequently, applying the same techniques to LLMs may not yield comparable effectiveness, as their activation distributions can differ significantly from those of VLMs due to the absence of multimodal integration. We plan to explore this direction in the future.
>
> **Q2: The paper lacks comparisons with state-of-the-art post-training quantization methods in terms of inference time, memory usage, and accuracy.**
>
> **Ans:** In our original manuscript (Table 2), we presented **accuracy comparisons** between QSVD and several leading post-training quantization methods, including Duquant [1] and QVLM [2]. Additionally, we extended our evaluation by comparing QSVD with **more recent PTQ approaches**, including MBQ [3] (targeting VLMs) and AffineQuant (designed for LLMs) [4]. The table below shows the evaluation results in terms of accuracy:
>
> | **Accuracy**     |         | **SQA**                     |                      |                      | **VizWiz**                  |                      |                      |
> |------------------|---------|-----------------------------|----------------------|----------------------|-----------------------------|----------------------|----------------------|
> |                  |         | MBQ                         | AffineQuant          | Ours QSVD            | MBQ                         | AffineQuant          | Ours QSVD            |
> | llava v1.5 7B     | W4A4    | 39.27                       | 46.23                | **55.16**            | 35.61                       | 42.63                | **52.05**            |
> |                  | W8A4    | 54.62                       | 59.45                | **65.61**            | 43.85                       | 46.22                | **52.18**            |
> | llava v1.5 13B    | W4A4    | 50.59                       | 58.55                | **65.82**            | 40.00                       | 45.56                | **56.82**            |
> |                  | W8A4    | 55.51                       | 67.83                | **70.12**            | 48.72                       | 50.25                | **53.20**            |
> | llava v1.6 7B     | W4A4    | 50.69                       | 53.72                | **59.67**            | 40.13                       | 49.90                | **52.00**            |
> |                  | W8A4    | 63.62                       | 65.87                | **66.10**            | 45.25                       | 52.75                | **53.72**            |
> | llava v1.6 13B    | W4A4    | 60.00                       | 61.24                | **63.61**            | 45.00                       | 49.11                | **54.27**            |
> |                  | W8A4    | 63.12                       | 68.97                | **70.43**            | 50.01                       | 56.41                | **58.52**            |
>
> We observe that QSVD consistently achieves comparable or superior **accuracy**, particularly under aggressive quantization settings such as W8A4 and W4A4.
>
> To provide a fair and concrete comparison with standard PTQ methods in **memory usage**, we conducted an experiment on one Transformer layer of LLaVA-v1.5-13B, focusing on the decoding stage. Specifically, we compare the following configurations:
>
> - **W8A8 PTQ:** Linear layers, KV cache and attention use standard INT8 quantization.
>
> - **QSVD-8bit:** Low-rank decomposition with int8 quantization, using our fused attention kernel optimized for compressed latent cache structure.
>
> For the W8A8 PTQ baseline, we adopt the per-tensor RTN (round-to-nearest)  quantization scheme, one of the simplest and most memory-efficient PTQ strategies, with no additional quantization parameters. This sets a conservative lower bound on memory usage. Most other SOTA PTQ methods introduce additional per-channel scales, zero-points, or outlier handling, which increase memory usage and inference time beyond RTN.
>
> We measured the peak memory usage in MB during decoding under varying sequence lengths (1024 and 2048) and batch sizes (16, 32, 160). The results are summarized below (Peak Memory Usage in MB):
>
> | Seq Len: 1024 | Memory |  Usage | (MB) |
> | --- | --- | --- | --- |
> | Batch Size | 16 | 32 | 160 |
> | W8A8 | 702.76 | 944.18 | 2875.52 |
> | QSVD-8bit | **545.97** | **633.14** | **1335.66** |
>
> | Seq Len: 2048 | Memory  | Usage | (MB) |
> | --- | --- | --- | --- |
> | Batch Size | 16 | 32 | 160 |
> | W8A8 | 943.51 | 1424.93 | 5276.27 |
> | QSVD-8bit | **626.97** | **824.12**| **2275.41** |
>
> As shown, under the same 8-bit precision, QSVD achieves significantly lower **peak memory usage** than standard W8A8 PTQ, especially under longer sequence lengths and larger batch sizes.
>
> Figure 4 in the paper shows the **inference latency** comparison between QSVD-8bit and W8A8. The same advantage is observed, QSVD-8bit achieves up to $1.4\times$ speedup over standard W8A8 PTQ during decoding. This improvement is due to both reduced memory traffic, as QSVD compresses the attention cache and lowers key/value memory reads, and fused kernel execution, which eliminates intermediate memory writes and enables more efficient computation by directly operating on the compressed representation.
>
> [1] Lin, Haokun, et al. "Duquant: Distributing outliers via dual transformation makes stronger quantized llms." Advances in Neural Information Processing Systems 37 (2024): 87766-87800.
>
> [2] Wang, Changyuan, et al. "Q-vlm: Post-training quantization for large vision-language models." Advances in Neural Information Processing Systems 37 (2024): 114553-114573.
>
> [3] Li, Shiyao, et al. "Mbq: Modality-balanced quantization for large vision-language models." Proceedings of the Computer Vision and Pattern Recognition Conference. 2025.
>
> [4] Ma, Yuexiao, et al. "Affinequant: Affine transformation quantization for large language models." arXiv preprint arXiv:2403.12544 (2024).
>
> **Q3: The figures need to be more refined. For example, Figure 2 alone is hard to interpret—the connections and meanings of the different subplots are unclear.**
>
> **Ans:** We thank the reviewer for the helpful feedback. To improve clarity and readability, we have clarified the connections and meanings as follows. We will also enhance the quality of Figure 2 in the final version of the paper. Specifically
>
> Subplot **(a)** presents the original QKV architecture in VLM without applying SVD.
>
> Subplot **(b)** illustrates the application of SVD separately to the weight matrices of $Q$ , $K$ and $V$ , where each of $W_{q}$ , $W_{k}$ , and $W_{v}$ is decomposed into a pair of down and up projection matrices.
>
> Subplot **(c)** shows our proposed QKV weight concatenation followed by the SVD process.
>
> Subplot **(d)** depicts the standard KV vectors computation process. At each decoding step, the input $X$ is multiplied with $W_{k}$ and $W_{v}$, and the resulting $C_{k}$ and $C_{v}$ are stored in memory for future use.
>
> Subplot **(e)** illustrates the recomputation process when SVD is applied separately to $W_{k}$ and $W_{v}$. During decoding, the intermediate input $X$ must be read from memory, and the $K$ and $V$ vectors are recomputed by multiplying $X$ with the down-projection matrices $W^{d}_k$  and  $W^{d}_v$, respectively.
>
> Subplot **(f)** demonstrates the storage cost in QSVD. During decoding, since $W_{q}$, $W_{k}$, and $W_{v}$ share a common down-projection matrix $W_{qkv}^{d}$ , the input $X$ is multiplied by $W_{qkv}^{d}$ to produce the intermediate data $C_{qkv}$, which is then stored in memory for later recomputation of the KV vectors.

---

> > ### Comment · Reviewer_HD9e · 2025-08-05
> > **Replying to Rebuttal by Authors**
> >
> > Thank you for the authors’ response. I feel that most of my concerns have been addressed. However, the explanation of how the proposed method specifically tackles the unique challenges of VLMs remains somewhat unclear. Therefore, I have changed my rating from borderline reject to borderline accept.

---

### Official Review · Reviewer_3Q9H · 2025-07-01

**Clarity:** 3
**Significance:** 3
**Originality:** 3
**Rating:** 5
**Confidence:** 3

**Summary:**

This paper proposes QSVD, a framework that leverages joint Singular Value Decomposition (SVD) of QKV weight matrices and quantization to reduce computational overhead and KV cache size in Vision-Language Models (VLMs). Key innovations include:
1. Joint QKV-SVD: Concatenating Q/K/V weight matrices for unified low-rank decomposition, reducing parameters and computation

2. Dynamic Rank Allocation: Singular value truncation based on impact on VLM accuracy (Eq. 10).



3. Low-Rank Activation Quantization: Orthogonal transformations to suppress activation outliers (Eq. 11-12).

Experiments show QSVD outperforms baselines (ASVD, SVD-LLM) on LLaVA and SmolVLM, reducing KV cache to 9.38% in W8A4 quantization with <1% accuracy drop on ScienceQA.

**Questions:**

1. Does the O(E²) cost of Eq. 10 affect real-time performance?

2. How does r selection adapt to long sequences (e.g., high-res images)?

3. Could visual-textual feature misalignment hinder joint SVD?

**Ethical Concerns:**

["NO or VERY MINOR ethics concerns only"]

**Final Justification:**

5

**Limitations:**

Yes

**Quality:**

3

**Strengths And Weaknesses:**

**Strengths:**
- Efficiency: Joint SVD+quantization reduces KV cache (η<sub>qsvd</sub> = *rL* vs. η<sub>fp</sub> = *2LE*) and computation (FLOPs ↓ when *r* < 0.75*E*).
- Novel Ranking Strategy: Global singular value importance scoring (Eq. 10) outperforms uniform/Fisher-based allocation (Table 3).
- Strong Validation: Tested on 5 VLMs and 3 benchmarks with W4A4 robustness (Table 2).

**Weaknesses:**
- Attention-Layer Limitation: No optimization for FFN layers (Sec. 5).

---

> ### Author Rebuttal · Authors · 2025-07-31
>
> Thank you for your insightful comments. Below, we summarize the key concerns and questions raised in your review and provide detailed responses to each point.
>
> **Q1: Attention-Layer Limitation: No optimization for FFN layers (Sec. 5).**
>
> **Ans:** As noted in earlier works [1-4], self attention layers, especially the key and value (KV) vectors, contribute significantly to the overall latency, energy consumption, and memory usage of large models and can severely limit inference speed. Therefore, optimizing the QKV matrices using SVD is critically important and can lead to substantial reductions in latency for large language models.
>
> In addition, as noted in Line 315 of the paper, quantization is applied to the FFN layers and already yields significant computational savings. When applying SVD to the FFN layers, we observed that FFN are highly sensitive to SVD, so we chose to use quantization alone, consistent with the observation of prior works [5–7]. Future research can further explore methods to better optimize the FFN layers.
>
>
> [1] Hooper, Coleman, et al. "Kvquant: Towards 10 million context length llm inference with kv cache quantization." *Advances in Neural Information Processing Systems* 37 (2024): 1270-1303.
>
> [2] Kim, Minsu, et al. "Oaken: Fast and Efficient LLM Serving with Online-Offline Hybrid KV Cache Quantization." *Proceedings of the 52nd Annual International Symposium on Computer Architecture*. 2025.
>
> [3] Kwon, Woosuk, et al. "Efficient memory management for large language model serving with pagedattention." *Proceedings of the 29th symposium on operating systems principles*. 2023.
>
> [4] Zhang, Zhenyu, et al. "H2o: Heavy-hitter oracle for efficient generative inference of large language models." *Advances in Neural Information Processing Systems* 36 (2023): 34661-34710.
>
> [5] Yuan, Zhihang, et al. "Asvd: Activation-aware singular value decomposition for compressing large language models." *arXiv preprint arXiv:2312.05821* (2023).
>
> [6] Chang, Chi-Chih, et al. "xkv: Cross-layer svd for kv-cache compression." *arXiv preprint arXiv:2503.18893* (2025).
>
> [7] Chang, Chi-Chih, et al. "Palu: Compressing kv-cache with low-rank projection." *arXiv preprint arXiv:2407.21118* (2024).
>
> **Q2: Does the $\mathcal{O}(E^2)$ cost of Eq. 10 affect real-time performance?**
>
> **Ans:** Thank you for the question. The $\mathcal{O}(E^2)$ cost in Equation 10 only applies during the **offline calibration phase**, where we compute importance scores to allocate ranks using a small calibration dataset. We benchmarked the complete QSVD pipeline for LLaVA-v1.5-13B on a single A100 80G GPU, using the same 256-sample calibration set as described in the paper. The total time is approximately 96 minutes. This is much shorter than the normal training time (204 GPU hours for LLaVA-v1.5-13B [1]). QSVD generates low-rank and quantized weights, which are stored and efficiently loaded during inference, in line with approaches such as AWQ [2] and GPTQ [3]. There is no need to recompute any part of the calibration during inference, making QSVD a one-time lightweight process with minimal runtime overhead.
>
> [1] Liu, Haotian, et al. "Improved baselines with visual instruction tuning." Proceedings of the IEEE/CVF conference on computer vision and pattern recognition. 2024.
>
> [2] Lin, Ji, et al. "Awq: Activation-aware weight quantization for on-device llm compression and acceleration." Proceedings of machine learning and systems 6 (2024): 87-100.
>
> [3] Frantar, Elias, et al. "Gptq: Accurate post-training quantization for generative pre-trained transformers." arXiv preprint arXiv:2210.17323 (2022).
>
> **Q3: How does r selection adapt to long sequences (e.g., high-res images)?**
>
> **Ans:** Thank you for the thoughtful comment. To evaluate the adaptability of our rank selection method to long sequences, we conduct experiments on the HRBench4K dataset [1], which consists of 4K resolution images, and use VLMEvalKit [2] for evaluation. We follow the same setup as the original paper and report the “Average All” accuracy metric in HRBench4K. Both LLaVA-v1.6-13B and LLaVA-v1.5-13B are evaluated under the QSVD-noQ setting and compared with ASVD and FP16 baselines. The results are shown below, where R1 and R2 follows the same definition as Equation 15 in the paper:
>
> **LLaVA-v1.6-13B**
>
> | Method     | Acc.   | Hw cost                    |   | Acc.   | Hw cost                   |    | Acc.   | Hw cost                     |  | Acc.   | Hw cost                       |
> |------------|--------|-----------------------------|--|--------|-----------------------------|--|--------|------------------------------|-|--------|-------------------------------|
> | ASVD       | 44.38% | R1: 63.30%; R2: 22.50%      |  | 44.12% | R1: 60.00%; R2: 20.00%       | | 43.12% | R1: 56.70%; R2: 17.50%     |   | 42.62% | R1: 53.30%; R2: 15.00%        ||
> | QSVD-noQ   | **44.88%** | R1: 60.00%; R2: 22.50%      |  | **44.12%** | R1: 53.30%; R2: 20.00%      |  | **43.88%** | R1: 46.70%; R2: 17.50%       | | **44.12%** | R1: 40.00%; R2: 15.00%        |
> | FP16       | 45.63% |                              |      |                             |       |                              |       |                              |
>
> **LLaVA-v1.5-13B**
>
> | Method     | Acc.   | Hw cost                      | | Acc.   | Hw cost                   |    | Acc.   | Hw cost                   |    | Acc.   | Hw cost                       |
> |------------|--------|-----------------------------|--|--------|------------------------------|-|--------|-------------------------------|----|----|-------------------------------|
> | ASVD       | 39.12% | R1: 63.30%; R2: 22.50%      |  | 38.62% | R1: 60.00%; R2: 20.00%      |  | 36.62% | R1: 56.70%; R2: 17.50%       | | 37.25% | R1: 53.30%; R2: 15.00%     |   |
> | QSVD-noQ   | **39.88%** | R1: 60.00%; R2: 22.50%    |   | **38.75%** | R1: 53.30%; R2: 20.00%      |  | **39.00%** | R1: 46.70%; R2: 17.50%  |  | **39.00%** | R1: 40.00%; R2: 15.00%      |  |
> | FP16       | 39.12% |                              |       |                              |       |                              |       |                              |
>
> As shown above, QSVD-noQ outperforms ASVD, and the overall trends observed on HRBench4K are consistent with those on ScienceQA-IMG and VizWiz (see Table 1 in our paper). This indicates that our rank selection method generalizes well to long sequences resulting from high-resolution images.
>
> [1] Wang, Wenbin, et al. "Divide, conquer and combine: A training-free framework for high-resolution image perception in multimodal large language models." Proceedings of the AAAI Conference on Artificial Intelligence. Vol. 39. No. 8. 2025.
>
> [2] Duan, Haodong, et al. "Vlmevalkit: An open-source toolkit for evaluating large multi-modality models." Proceedings of the 32nd ACM international conference on multimedia. 2024.
>
> **Q4: Could visual-textual feature misalignment hinder joint SVD?**
>
> **Ans:** Thank you for this insightful question. Visual-textual feature misalignment is a well-known challenge in VLMs and can significantly impact overall model accuracy. However, it does not directly hinder the effectiveness of our proposed joint SVD approach in QSVD, as QSVD is designed to maintain the original VLM's performance even under feature misalignment, while simultaneously enhancing computational efficiency. Furthermore, by compressing Q, K, and V jointly, QSVD inherently smooths parameter variations across modalities. This may even help mitigate alignment-induced instability by regularizing the QKV projection layer and encouraging a more cohesive representation space. Additionally, the evaluation in Section 4 shows consistent performance across diverse VLMs (e.g., SmolVLM, LLaVA) and datasets (e.g., ScienceQA, VizWiz), despite differences in visual-textual alignment quality. This suggests that joint SVD is robust to potential modality misalignments at the embedding level.
>
> Finally, we simulate the effect of feature misalignment by injecting Gaussian noise into the visual features generated by the visual encoder. Specifically, we perturb the output of the projection layer with Gaussian noise at varying levels, using a standard deviation of 0.1. The corrupted visual features are then concatenated with the textual embeddings and passed through the subsequent layers for processing. Following the experimental setup in the main paper (Table 4), we use the LLaVA-v1.5-7B model and evaluate performance on the ScienceQA benchmark.
>
> The results, shown in the table below, indicate that QSVD exhibits a certain degree of robustness to feature misalignment, performing noticeably better than ASVD under the same conditions. This observation suggests the potential resilience of joint SVD to modality noise, which warrants further investigation in future work.
>
>
> | Setting            | FP16-ASVD | FP16-QSVD-noQ | W4A4-QSVD |
> |--------------------|-----------|---------------|-----------|
> | W/o noise in feature | **66.34** | **68.46**      | **55.16** |
> | W/ noise in feature  | 46.24     | 50.02          | 48.29     |

---

> > ### Comment · Reviewer_3Q9H · 2025-08-05
> > **Good rebuttal**
> >
> > 5->5. I wish to see all these comments and supplementary experiments can be posed on the camera-ready version.

---

### Official Review · Reviewer_bUxq · 2025-07-02

**Clarity:** 3
**Significance:** 2
**Originality:** 3
**Rating:** 5
**Confidence:** 3

**Summary:**

This paper introduces a QSVD method that combines quantization and SVD. It cocnats QKV weight matrix together and performs a low-rank SVD on it. Then it introduces a cross‑layer, importance‑based singular‑value ranking scheme for adaptive rank allocation. This method can reduce the GPU memory usage and computation cost, and experiments on five VLMs across ScienceQA, VizWiz, and SEED‑Bench demonstrate that QSVD outperforms SVD‑only and quantization‑only baselines.

**Questions:**

1. Analyze why we can get better results from SVD on concated QKV. Either theory or experiment is encouraged.
2. What is the beta obtained on the calibration dataset? Can authors study the beta values on different datasets? This ablation is meaningful for practical applications.
3. Experiment on GPU memory usage between different methods is encouraged.

**Ethical Concerns:**

["NO or VERY MINOR ethics concerns only"]

**Final Justification:**

Post rebuttal, my concerns about method explanations, ablation experiments and GPU consumption have been addressed. Thus I am leaning to accept the paper.

**Limitations:**

Yes

**Quality:**

3

**Strengths And Weaknesses:**

1. The joint QKV SVD formulation is clearly derived, with precise comparisons against single matrix SVD and analysis of parameter/FLOPs/cache reductions. The results support the method proposed.
2. The figures are intuitive. A small drawback is that the order of the sub-graphs is inconsistent with the order in the text. 2. The parameter beta is only used in Quantization, so it’s confusing in Section 3.1 without explanation.
3. QSVD works on different models and datasets, but why do we need to concatenate the three matrices together? Is it just to reduce cache overhead? Or is there a deeper reason for this design? A deeper analysis is encouraged.
4. While prior work apply SVD to Q, K, V separately, this paper’s joint factorization over the concatenated matrix is novel (despite the lack of a deeper motivation)
5. The knockout approach to measuring the importance of singular values is novel.

---

> ### Author Rebuttal · Authors · 2025-07-31
>
> Thank you for your insightful comments. We have summarized weakness and questions raised in your review below and provided detailed responses to each.
>
> **Q1: The figures are intuitive. A small drawback is that the order of the sub-graphs is inconsistent with the order in the text.**
>
> **Ans:** We will clarify the figure in the final version of the paper.  And please see **Reviewer** **HD9e Q3**
>
>
> **Q2: The parameter beta is only used in Quantization, so it’s confusing in Section 3.1 without explanation.**
>
> **Ans:** The parameter $\beta$, although introduced earlier in Section 3.1, does not affect the outcome of the SVD itself. Specifically, when SVD is applied to a weight matrix $W$, the process can be expressed as:
>
> $W = U \Sigma V^T = (U \Sigma^\beta)(\Sigma^{1-\beta} V^T) = W_{d}W_{u}$
>
> The choice of $\beta$ does not alter the reconstruction result of the SVD. We note this here to support the discussion of $\beta$ in Section 3.3.
>
> However, once quantization is introduced, the value of $\beta$ becomes impactful on the final accuracy result, as it influences the magnitudes of $W_d$, $W_u$, and the intermediate product $C_{qkv} = XW_d$, and their corresponding quantized representations. Therefore, learning an appropriate value for $\beta$ is crucial for optimizing quantization performance.
>
> **Q3: QSVD works on different models and datasets, but why do we need to concatenate the three matrices together? Is it just to reduce cache overhead? Or is there a deeper reason for this design? Analyze why we can get better results from SVD on concated QKV. Either theory or experiment is encouraged.**
>
> **Ans:** In addition to reducing the KV cache size, another potential advantage of applying SVD to the concatenated QKV weights is the improved flexibility in assigning singular values compared to applying SVD to each of the Q, K, and V weights separately. For example, given a total rank budget of 9, performing SVD on Q, K, and V individually would require a fixed allocation (such as 3, 3, and 3), which may result in truncating many important singular values and lead to suboptimal performance. In contrast, applying SVD to the concatenated QKV weights allows for a global comparison of singular values across all three matrices. This enables more adaptive and effective rank allocation under the same total budget, potentially yielding better overall performance.
>
> To illustrate the advantages of QKV concatenation over separate QKV decomposition for SVD, we provide a quantitative comparison below, summarized from Table 1 and Table 3 of our paper. The comparison uses LLaVA-v1.5-13B under different preserved parameters across both settings, the definitions of $R_{1}$ and $R_{2}$ follow equation 15 in the paper.
>
> | Source | SVD Type | Acc. | Hw cost | Acc. | Hw cost | Acc. | Hw cost | Acc. | Hw cost |
> |--------|----------|------|-----------------------------|------|-----------------------------|------|-----------------------------|------|-----------------------------|
> | From Table 1, LLaVA-v1.5 13B, ASVD | Separately SVD | 64.70% | R1: 63.30%; R2: 22.50% | 56.92% | R1: 60.00%; R2: 20.00% | 46.50% | R1: 56.70%; R2: 17.50% | 42.79% | R1: 53.30%; R2: 15.00% |
> | From Table 3, LLaVA-v1.5 13B, UR | Joint SVD       | 71.84% | R1: 60.00%; R2: 22.50%  | 70.40% | R1: 53.30%; R2: 20.00% | 70.40% | R1: 46.70%; R2: 17.50% | 67.72% | R1: 40.00%; R2: 15.00%  |
>
> As the table shows, joint SVD improves accuracy by more than **7.14%** under the same compression budget. This trend holds consistently across various preserved ratios, R1 and R2 are defined in Equation 15.
>
> Finally, concatenating the Q, K, and V weights allows for a shared down projection, which significantly reduces the KV cache size during inference. As shown in Figure 2 and discussed in Section 3.1, our design stores a single intermediate matrix $C_{{qkv}}$ instead of computing and caching separate key and value projections for each $W_{{k}}$ and $W_{{v}}$. This results in approximately a twofold reduction in KV cache size compared to traditional approaches.
>
> **Q4: What is the beta obtained on the calibration dataset? Can authors study the beta values on different datasets? This ablation is meaningful for practical applications.**
>
> **Ans:**  In our paper. We follow the convention of QVLM [1] to use samples from ScienceQA training set as our calibration dataset, and the resulting beta is around 0.12. In contrast, MBQ [2] uses the COCO [3] dataset for its calibration process. To align with this, we extended our beta learning experiments using COCO data and observed a similar optimal beta value of approximately 0.10. This consistency indicates that beta remains stable across different calibration datasets, which significantly enhances the practicality of QSVD, since a beta value of 0.1 can be reliably applied across various datasets.
>
> [1] Wang, Changyuan, et al. "Q-vlm: Post-training quantization for large vision-language models." NeurIPS, 2024.
>
> [2] Li, Shiyao, et al. "Mbq: Modality-balanced quantization for large vision-language models." Proceedings of the Computer Vision and Pattern Recognition Conference. 2025.
>
> [3] Lin, Tsung-Yi, et al. "Microsoft coco: Common objects in context." European conference on computer vision. Cham: Springer International Publishing, 2014.
>
> **Q5: Experiment on GPU memory usage between different methods is encouraged.**
>
> **Ans:** To provide a fair and concrete comparison with standard PTQ methods in memory cost, we conducted an experiment on one transformer layer of LLaVA-v1.5-13B, focusing on the decoding stage. Specifically, we compare the following approaches:
>
> - **W8A8 PTQ:** Linear layers, KV cache and attention use standard INT8 quantization.
>
> - **QSVD-8bit:** Low-rank decomposition with int8 quantization, using our fused attention kernel optimized for compressed latent cache structure.
>
> For the W8A8 PTQ baseline, we adopt the per-tensor RTN (round to nearest) quantization scheme, one of the simplest and most memory-efficient PTQ strategies, with no additional quantization parameters. This sets a conservative lower bound on memory usage. Most other recent PTQ methods introduce additional per-channel scales, zero-points, or outlier handling, which increase memory usage beyond RTN.
>
> We measured the peak memory usage in MB during decoding under varying sequence lengths (1024 and 2048) and batch sizes (16, 32, 160). The results are summarized below (Peak Memory Usage in MB):
>
> | Seq Len: 1024 | Memory  | Usage | (MB) |
> | --- | --- | --- | --- |
> | Batch Size | 16 | 32 | 160 |
> | W8A8 | 702.76 | 944.18 | 2875.52 |
> | QSVD-8bit | **545.97** | **633.14** | **1335.66**|
>
> | Seq Len: 2048 | Memory  | Usage | (MB) |
> | --- | --- | --- | --- |
> | Batch Size | 16 | 32 | 160 |
> | W8A8 | 943.51 | 1424.93 | 5276.27 |
> | QSVD-8bit | **626.97** | **824.12** | **2275.41** |
>
> As shown, under the same 8-bit precision, QSVD achieves significantly lower peak memory usage than standard W8A8 PTQ, especially under longer sequence lengths and larger batch sizes.

---

> > ### Comment · Reviewer_bUxq · 2025-08-04
> >
> > Thanks for your thoughtful response. My concerns have been addressed. I will raise my score.

---

### Official Review · Reviewer_Ue5o · 2025-07-03

**Clarity:** 3
**Significance:** 3
**Originality:** 3
**Rating:** 5
**Confidence:** 4

**Summary:**

This paper proposes QSVD, a unified framework for compressing Vision-Language Models by applying joint Singular Value Decomposition to the combined query, key, and value weight matrices, alongside post-training quantization of weights and activations. The goal is to reduce memory, computational cost, and KV cache size without degrading model accuracy.

**Questions:**

- Can we replace post-training quantization by quantization-aware training and further improve the performance?
- What about the training time? Does QSVD add heavy additional cost compared with normal training?

**Ethical Concerns:**

["NO or VERY MINOR ethics concerns only"]

**Final Justification:**

See comments

**Limitations:**

Yes

**Quality:**

3

**Strengths And Weaknesses:**

Strengths:
- This paper proposes a novel compression paradigm that effectively combines SVD and PTQ.
- The buffered mechanism and auto-regressive decoding work well together.
- The figures well demonstrate the method design.
- Extensive experiments on popular VLMs show superior performance over existing works.

Weaknesses:
- The method's effectiveness is quite sensitive to $\beta$.  And $\beta$ is determined by a very small calibration set. I'm not sure if the generalization ability of such pattern is enough.
- The method can only be used on QKV matrices and can not be used on other layers like FC layers, restricting its compression capacity.

---

> ### Author Rebuttal · Authors · 2025-07-31
>
> Thank you for your insightful comments. Below, we summarize the weaknesses and questions raised in your review and provide detailed responses to each.
>
> **Q1: The method's effectiveness is quite sensitive to $\beta$ . And $\beta$ is determined by a very small calibration set. I'm not sure if the generalization ability of such pattern is enough.**
>
> **Ans:** We explored the beta learning experiments on different sample size of 64, 128, 256 and 512. Furthermore, to show the effectiveness of beta learning, we also show the QSVD performances when fixing beta to 0.2, 0.4, 0.8 and 1. The results for W4A4 and W8A4 are shown in table below:
> | LLaVA v1.5 7B |  |  |  |  |  |  |
> | --- | --- | --- | --- | --- | --- | --- |
> |  |$\beta$ | QSVD| $\beta$=0.2 | $\beta$=0.4 | $\beta$=0.8 | $\beta$=1 |
> | n=64 | W4A4 | **52.31** | 48.44 | 42.24 | 39.32 | 30.05 |
> |  | W8A4 | **64.46** | 63.36 | 61.38 | 58.40 | 49.43 |
> | n=128 | W4A4 | **46.85** | 42.54 | 35.30 | 36.76 | 30.05 |
> |  | W8A4 | **65.74** | 62.02 | 63.96 | 61.38 | 47.79 |
> | n=256 (QSVD default setting) | W4A4 | **55.16** | 54.79 | 42.37 | 49.22 | 40.45 |
> |  | W8A4 | **65.61** | 64.06 | 64.35 | 60.27 | 58.58 |
> | n=512 | W4A4 | **41.65** | 33.12 | 26.56 | 17.09 | 10.55 |
> |  | W8A4 | **63.91** | 62.17 | 63.71 | 59.44 | 13.93 |
>
> The table shows that the selection of $\beta$ matches in different sample size settings, and this learning $\beta$ method generalizes well over different calibration sizes.
>
> Moreover, it is standard in post-training quantization and low-rank compression literature to use small calibration sets to estimate sensitive hyperparameters. For example: AWQ [1] Section 5.3 Figure 8(a) use 8-256 samples to determine the scaling factor exponent. ASVD [2] Section 4.1 use only 64 samples of calibration dataset in all their experiments and hyperparameter search.
>
> [1] Lin, Ji, et al. "Awq: Activation-aware weight quantization for on-device llm compression and acceleration." Proceedings of machine learning and systems 6 (2024): 87-100.
>
> [2] Yuan, Zhihang, et al. "Asvd: Activation-aware singular value decomposition for compressing large language models." arXiv preprint arXiv:2312.05821 (2023).
>
> **Q2: The method can only be used on QKV matrices and can not be used on other layers like FC layers, restricting its compression capacity.**
>
> **Ans:** As noted in earlier works [1-4], self attention layers, particularly the KV vectors, contribute significantly to the overall latency, energy consumption, and memory usage of large models, and can severely limit inference speed. Therefore, optimizing the QKV matrices using singular value decomposition is critically important and can lead to substantial reductions in latency for large language models.
>
> In addition, as mentioned in Line 315 of our paper, quantization is applied to the FC layers within the feedforward network (FFN), as well as the output FC layers within the self attention mechanism. While SVD can also be applied to these FC layers, we find them to be highly sensitive to such decomposition. Applying quantization alone is sufficient to compress these layers effectively. This design choice aligns with prior work [5, 6]. Future research may explore more advanced strategies to further optimize the FFN components.
>
> [1] Hooper, Coleman, et al. "Kvquant: Towards 10 million context length llm inference with kv cache quantization." *Advances in Neural Information Processing Systems* 37 (2024): 1270-1303.
>
> [2] Kim, Minsu, et al. "Oaken: Fast and Efficient LLM Serving with Online-Offline Hybrid KV Cache Quantization." *Proceedings of the 52nd Annual International Symposium on Computer Architecture*. 2025.
>
> [3] Kwon, Woosuk, et al. "Efficient memory management for large language model serving with pagedattention." *Proceedings of the 29th symposium on operating systems principles*. 2023.
>
> [4] Zhang, Zhenyu, et al. "H2o: Heavy-hitter oracle for efficient generative inference of large language models." *Advances in Neural Information Processing Systems* 36 (2023): 34661-34710.
>
> [5] Yuan, Zhihang, et al. "Asvd: Activation-aware singular value decomposition for compressing large language models." *arXiv preprint arXiv:2312.05821* (2023).
>
> [6] Chang, Chi-Chih, et al. "xkv: Cross-layer svd for kv-cache compression." *arXiv preprint arXiv:2503.18893* (2025).
>
> [7] Chang, Chi-Chih, et al. "Palu: Compressing kv-cache with low-rank projection." *arXiv preprint arXiv:2407.21118* (2024).
>
> **Q3: Can we replace post-training quantization by quantization-aware training and further improve the performance?**
>
> **Ans:** Yes, incorporating quantization-aware training (QAT) has the potential to further improve quantization performance by allowing model weights to adapt to low-precision constraints during training. However, this approach incurs significant training cost, as backpropagation must be performed across the entire vision-language model, resulting in substantial computational overhead and increased training latency. In contrast, our primary goal in this work is to develop a highly effective **post-training quantization (PTQ)** method that is both efficient and practical.
>
> SpinQuant [1] demonstrates that end-to-end QAT can yield superior results compared to QuaRot under aggressive settings (e.g., W4A4), achieving up to **28.6-percent improvements** (see SpinQuant Table 5, Section 4.3.4). While impressive, this comes at the cost of additional training and online adaptation. Our method, like QuaRot, is **fully PTQ and no E2E training**, and we believe that with modest QAT integration (e.g., low-rank-aware finetuning or selective stage-wise updates), **QSVD has the potential have an even better performance**, which we leave as promising future work.
>
> [1] Liu, Zechun, et al. "Spinquant: Llm quantization with learned rotations." arXiv preprint arXiv:2405.16406 (2024).
>
> **Q4: What about the training time? Does QSVD add heavy additional cost compared with normal training?**
>
> **Ans:** Thank you for the question. QSVD is designed as a post-training compression method and does not require full model retraining. The additional computation involved comes from three components: (1) low-rank SVD decomposition of the QKV matrices, (2) importance score–based rank allocation, and (3) calibration-based tuning of the quantization parameter $\beta$ and the quantization process. The latency overhead introduced by QSVD is negligible compared to standard training procedures. Furthermore, it introduces no additional cost during inference.
>
> To study the detailed training cost, QSVD introduces only a lightweight calibration and tuning stage, rather than full fine-tuning. For a fair comparison, we benchmarked the complete QSVD pipeline for LLaVA-v1.5-13B on a single A100 80G GPU, using the same 256-sample calibration set as described in the paper. The total time is approximately 96 minutes. The detailed breakdown is:
>
> | Step | Time |
> | --- | --- |
> | Input calibration | 1 min |
> | SVD (on all layers) | 2 min 15 s |
> | Gradient collection & rank allocation | 10 min 12 s |
> | SVD results Fusion | 30 s |
> | $\beta$ tuning & quantization (fake quant) | 82 min |
> | Total Calibration Time | **96 min** |
>
> In comparison, the typical training cost of a vision-language model is approximately **204 GPU hours** [1] on A100, making the computational overhead of QSVD negligible relative to full model training.
>
> [1] Liu, Haotian, et al. "Improved baselines with visual instruction tuning." Proceedings of the IEEE/CVF conference on computer vision and pattern recognition. 2024.

---

> > ### Comment · Reviewer_Ue5o · 2025-08-07
> >
> > Thanks for the rebuttal. My concerns are mostly solved. I think this paper is technically solid and worth being spread. I keep my initial recommendation "accept".

---

### Comment · Area_Chair_1aQ8 · 2025-08-04
**Please check authors' rebuttal**

Dear reviewers,

Thank you for your efforts. The authors have provided a detailed response. Please check if any concerns remain and engage in discussion with authors.

Best,
AC

---

> ### Comment · Reviewer_3Q9H · 2025-08-06
>
> good

---

### Note · Authors · 2025-08-12

We sincerely thank all reviewers for their constructive feedback, thoughtful questions, and recognition of our work. The discussions and clarifications during the rebuttal period have been invaluable in refining both our technical presentation and our understanding of the broader impact of this research. We will incorporate the clarifications, supplementary experiments, and discussions raised in the rebuttal into the camera-ready version to further strengthen the paper. We deeply appreciate the time and effort each reviewer has invested in evaluating our work.

---

### Decision · Program_Chairs · 2025-09-17

**Decision:**

Accept (spotlight)

**Comment:**

Driven by the high computation cost and memory footprint of vision language models,  the authors propose a method to reduce the KV cash size. The authors suggest to concatenate the Key, Query and Value matrices and perform a Singular Value Decomposition with an efficient rank allocation.

Strengths

All reviewers found this interesting and novel.

The paper demonstrated strong performance.

Through experiments across various models.

Weakness

The main limitation pointed by the reviewers is the applicability to the attention layers only, however the authors showed that the work can be combined with FC quantization.